# Rethinking the Backward Propagation for Adversarial Transferability

**Xiaosen Wang[1]\*, Kangheng Tong[2]\*, Kun He[2]†**
[1]Huawei Singular Security Lab
[2]School of Computer Science and Technology, Huazhong University of Science and Technology
{xiaosen,tongkangheng,brooklet60}@hust.edu.cn

## Abstract

Transfer-based attacks generate adversarial examples on the surrogate model, which can mislead other black-box models without access, making it promising to attack real-world applications. Recently, several works have been proposed to boost adversarial transferability, in which the surrogate model is usually overlooked. In this work, we identify that non-linear layers (*e.g.* ReLU, max-pooling, *etc*.) truncate the gradient during backward propagation, making the gradient *w.r.t.* input image imprecise to the loss function. We hypothesize and empirically validate that such truncation undermines the transferability of adversarial examples. Based on these findings, we propose a novel method called Backward Propagation Attack (BPA) to increase the relevance between the gradient *w.r.t.* input image and loss function so as to generate adversarial examples with higher transferability. Specifically, BPA adopts a non-monotonic function as the derivative of ReLU and incorporates softmax with temperature to smooth the derivative of max-pooling, thereby mitigating the information loss during the backward propagation of gradients. Empirical results on the ImageNet dataset demonstrate that not only does our method substantially boost the adversarial transferability, but it is also general to existing transfer-based attacks. Code is available at `https://github.com/Trustworthy-AI-Group/RPA`.

## 1 Introduction

Deep Neural Networks (DNNs) have been widely applied in various domains, such as image recognition [37, 13, 16], object detection [34], face verification [52, 41], *etc*. However, their susceptibility to adversarial examples [39, 10], which are carefully crafted by adding imperceptible perturbations to natural examples, has raised significant concerns regarding their security. In recent years, the generation of adversarial examples, *aka* adversarial attacks, has garnered increasing attention [31, 21, 6, 55, 44, 24] in the research community. Notably, there has been a significant advancement in the efficiency and applicability of adversarial attacks [19, 1, 47, 6, 57, 59, 51], making them increasingly viable in real-world scenarios.

By exploiting the transferability of adversarial examples across different models [29], transfer-based attacks generate adversarial examples on the surrogate model to fool the target models [6, 57, 12, 43, 58]. Unlike other types of attacks [2, 19, 1, 47], transfer-based attacks do not require direct access to the victim models, making them particularly applicable for attacking online interfaces. Consequently, transfer-based attacks have emerged as a prominent branch of adversarial attacks. However, it is worth noting that the early white-box attacks [10, 31, 21] often exhibit poor transferability despite demonstrating superior performance within the white-box setting.

---

\*The first two authors contribute equally.
†Corresponding author.

To this end, different techniques have been proposed to enhance adversarial transferability, such as momentum-based attacks [6, 27, 43, 46, 9, 59], input transformations [57, 7, 45, 30, 8, 48, 42], advanced objective functions [17, 54, 49], and model-related attacks [23, 53, 12, 50]. Among these techniques, model-related attacks are particularly valuable due to their ability to exploit the characteristics of surrogate models. Model-related attacks offer a unique perspective on adversarial attacks by leveraging the knowledge gained from surrogate models, which can also shed new light on the design of more robust models. In despite of their potential significance, model-related attacks have been somewhat overlooked compared to other types of transfer-based attacks.

Since transfer-based attacks mainly design various gradient ascend methods to generate adversarial examples on the surrogate model, in this work, we first revisit the backward propagation procedure. We find that non-linear layers (*e.g.*, activation function, max-pooling, *etc.*) often truncate the gradient of loss *w.r.t.* the feature map, which diminishes the relevance of the gradient between the loss and input image. We assume and empirically validate that such gradient truncation undermines the adversarial transferability. Based on this finding, we propose Backward Propagation Attack (BPA), which modifies the calculation for the derivative of ReLU activation function and max-pooling layers during the backward propagation process. With these modifications, BPA mitigates the negative impact of gradient truncation and improves the transferability of adversarial attacks.

Our main contribution can be summarized as follows:

- To our knowledge, this is the first work that proposes and empirically validates the detrimental effect of gradient truncation on adversarial transferability. This finding sheds new light on improving adversarial transferability and might provide new directions to boost the model robustness.
- We propose a model-related attack called BPA, that adopts a non-monotonic function as the derivative of the ReLU activation function and incorporates softmax with temperature to calculate the derivative of max-pooling. With these modifications, BPA mitigates the negative impact of gradient truncation and enhances the relevance of gradient between the loss function and the input.
- Extensive experiments on ImageNet dataset demonstrate that BPA could significantly boost various untargeted and targeted transfer-based attacks and outperform the baselines with a substantial margin, emphasizing the effectiveness and superiority of our proposed approach.

## 2 Related Work

In this section, we provide a brief overview of the existing adversarial attacks and defenses.

### 2.1 Adversarial Attacks

Existing adversarial attacks can be categorized into two groups based on the access to target model, namely white-box attacks and black-box attacks. In the white-box setting [10, 33, 31, 2], attackers have complete access to the structure and parameters of the target model. In the black-box setting, the attacker has limited or no information about the target model, making it applicable in the physical world. Black-box attacks can be further grouped into three classes, *i.e.*, score-based attacks [4, 19], query-based attacks [3, 22, 47], and transfer-based attacks [6, 57, 43]. Among the three types of black-box attacks, transfer-based attacks generate adversarial examples on the surrogate model without accessing the target model, drawing increasing interest recently.

Since MI-FGSM [6] integrates momentum into I-FGSM [21] to stabilize the update direction and achieve improved transferability, various momentum-based attacks have been proposed to generate transferable adversarial examples. For instance, NI-FGSM [27] leverages Nesterov Accelerated Gradient for better transferability. VMI-FGSM [43] refines the current gradient using the gradient variance from the previous iteration, resulting in more stable updates. EMI-FGSM [46] enhances the momentum by averaging the gradient of several data points sampled in the previous gradient direction.

On the other hand, input transformations that modify the input image prior to gradient calculation have proven highly effective in enhancing adversarial transferability, such as DIM [56], TIM [7], SIM [27], Admix [45], SSA [30] and so on. Among these attacks, Admix introduces a small segment of an image from different categories, while SSA applies frequency domain transformations to the input image, both of which have demonstrated superior performance in generating transferable adversarial examples.

Several studies have explored the utilization of more sophisticated objective functions to enhance transferability in adversarial attacks. ILA [17] employs fine-tuning techniques to increase the similarity of feature differences between the original or current adversarial example and a benign sample. ATA [54] maximizes the disparity of attention maps between a benign sample and an adversarial example. FIA [49] minimizes a weighted feature map in an intermediate layer to disrupt significant object-aware features.

A few works have emphasized the significance of the surrogate model in generating highly transferable adversarial examples. Ghost network [23] attacks a set of ghost networks generated by densely applying dropout at the intermediate features. On the other hand, another line of work focuses on the gradient during backward propagation. SGM [54] adjusts the decay factor to incorporate more gradients from the skip connections of ResNet to generate more transferable adversarial examples. LinBP [12] performs backward propagation in a more linear fashion by setting the gradient of ReLU as a constant of 1 and scaling the gradient of residual blocks. In this work, we find that the gradient truncation introduced by non-linear layers undermines the transferability and modify the backward propagation so as to generate more transferable adversarial examples.

## 2.2 Adversarial Defenses

The existence of adversarial examples poses a significant security threat to deep neural networks (DNNs). To mitigate this impact, researchers have proposed various methods, among which adversarial training has emerged as a widely used and effective approach [10, 21, 31]. By augmenting the training data with adversarial examples, this method enhances the robustness of trained models against adversarial attacks. For instance, Tramèr et al. [40] introduce ensemble adversarial training, a technique that generates adversarial examples using multiple models simultaneously, which shows superior performance against transfer-based attacks.

Although adversarial training is effective, it comes with high training costs, particularly for large-scale datasets and complex networks. Consequently, researchers have proposed innovative defense methods as alternatives. Guo et al. [11] utilize various input transformations such as JPEG compression and total variance minimization to eliminate adversarial perturbations from input images. Xie et al. [56] mitigate adversarial effects through random resizing and padding of input images. Liao et al. [25] propose training a high-level representation denoiser (HGD) specifically designed to purify input images. Nasser [32] introduce a neural representation purifier (NRP) by a self-supervised adversarial training mechanism to purify the input sample. Various certified defenses aim to provide a verified guarantee in a specific radius, such as randomized smoothing (RS) [5].

## 3 Methodology

In this section, we analyze the backward propagation procedure and identify that the gradient truncation introduced by non-linear layers undermines the adversarial transferability. Based on this finding, we propose Backward Propagation Attack (BPA) to mitigate such negative effect and gain more transferable adversarial examples.

### 3.1 Backward Propagation for Adversarial Transferability

Given an input image $x$ with ground-truth label $y$, a classifier $f$ with $l$ successive layers (*e.g.*, $z_{i+1} = \phi_i(f_i(z_i))$, $z_0 = x$) predicts the label $f(x) = f_l(z_l) = y$ with high probability. Here $\phi(\cdot)$ is a non-linear activation function (*e.g.*, ReLU) or identity function if there is no activation function after $i$-th layer $f_i$. The attacker aims to find an adversarial example $x^{adv}$ adhering the constraint of $\|x^{adv} - x\|_p \leq \epsilon$, but resulting in $f(x^{adv}) \neq f(x) = y$ for untargeted attack and $f(x^{adv}) = y_t$ for targeted attack. Here $\epsilon$ is the maximum perturbation magnitude, $y_t$ is the target label, and $\| \cdot \|_p$ denotes the $p$-norm distance. For brevity, the following description will focus on non-targeted attacks with $p = \infty$. Let $J(x, y; \theta)$ denote the loss function of classifier $f$ (*e.g.*, the cross-entropy loss). Existing white-box attacks often solve the following constrained maximization problem using the gradient $\nabla_x J(x, y; \theta)$:

$$x^{adv} = \operatorname*{argmax}_{\|x'-x\|_p \leq \epsilon} J(x', y; \theta). \tag{1}$$

Based on the chain rule, we can calculate the gradient as follows:

$$\nabla_x J(x, y; \theta) = \frac{\partial J(x, y; \theta)}{\partial f_l(z_l)} \left( \prod_{i=k+1}^{l} \frac{\partial \phi_i(f_i(z_i))}{\partial z_i} \right) \frac{\partial z_{k+1}}{\partial z_k} \frac{\partial z_k}{\partial x}, \tag{2}$$

where $0 < k < l$ is the index of an arbitrary layer. Without loss of generality, we explore the backward propagation when passing the $k$-th layer as follows:

- **A fully connected or convolutional layer followed by a non-linear activation function**. Taking ReLU activation (*i.e.*, $\phi_k$) for example, the $j$-th element in the gradient *w.r.t.* the $k$-th feature, $[\frac{\partial z_{k+1}}{\partial z_k}]_j$, will be one if $z_{k,j} > 0$ and otherwise, $[\frac{\partial z_{k+1}}{\partial z_k}]_j$ will be zero. These zero gradients in $\frac{\partial z_{k+1}}{\partial z_k}$ can lead to the truncation of gradient of the loss function $\frac{\partial J(x,y;\theta)}{\partial z_k}$ *w.r.t.* the input image. As a result, the gradient is effectively limited or weakened to some extent.
- **Max-pooling layer**. As shown in Fig. 1, max-pooling calculates the maximum value (orange block) within a specific patch. Hence, the derivative $\frac{\partial z_{k+1}}{\partial z_k}$ will be a binary matrix, containing only ones at locations corresponding to the orange blocks. In this case, approximately $3/4$ of the elements in the given sample will be zeros. This means that max-pooling tends to discard a significant portion of the gradient information contained in $\frac{\partial z_k}{\partial x}$, resulting in a truncated gradient.

The truncation of gradient caused by non-linear layers (*e.g.*, activation function, max-pooling) can limit or dampen the flow of gradients during backward propagation, which decays the relevance among the gradient between the loss and input. Considering that many existing attacks rely on maximizing the loss by leveraging the gradient information, we make the following assumption:

**Assumption 1** *The truncation of gradient $\nabla_x J(x, y; \theta)$ introduced by non-linear layers in the backward propagation process decays the adversarial transferability.*

To validate Assumption 1, we conduct several experiments using FGSM, I-FGSM and MI-FGSM. The detailed experimental settings are summarized in Sec. 4.1.

| 0.1 | -0.2 | 1.9 | 1.4 |
| 0.0 | -0.5 | 2.3 | 0.7 |
| -0.4 | 0.9 | 1.0 | -2.0 |
| 0.7 | 0.6 | 0.5 | 1.7 |

Figure 1: A max-pooling layer with $2 \times 2$ kernel size and stride $s = 2$ on a $4 \times 4$ feature map in the forward propagation.

- **Randomly mask the gradient**. To investigate the impact of gradient truncation on adversarial transferability, we introduce a random masking operation to increase the probability of gradient truncation between stage 3 and stage 2 of ResNet-50. Fig. 2a illustrates the attack performance with various mask probabilities. As the mask probability increases, more zeros appear in the derivative, indicating a higher degree of gradient truncation. Consequently, the larger truncation probability renders the gradient less relevant to the loss function, decreasing the attack performance of the three evaluated methods. These findings validate our hypothesis that the truncation of gradient negatively impacts adversarial transferability and highlight the importance of preserving gradient information to maintain the effectiveness of adversarial attacks across various models.
- **Recover the gradient of ReLU or max-pooling layers**. In contrast, it is expected that mitigating the truncation of gradient can improve the adversarial transferability. To explore this, we randomly replaced the zeros in the derivative of ReLU or max-pooling operations with ones, using various replacement probabilities. a) In Fig. 2b, as the probability of replacement increases, fewer gradients are truncated across ReLU, resulting in improved adversarial transferability on all the three attacks. Notably, these attacks achieve their best performance when the derivative consists entirely of ones, which aligns with LinBP [12]. b) As illustrated in Fig. 2c, when the ratio of ones in the derivative of max-pooling increases (*i.e.*, the replacement probability increases), the attack performance initially improves, reaching a peak around $0.3$. Subsequently, the attack performance gradually decreases but remains superior to vanilla backward propagation. This highlights that we need a more suitable approximation for the derivative calculation of max-pooling, which is detailed in Sec. 3.2. These results suggest that decreasing the probability of gradient truncation in max-pooling is beneficial for enhancing adversarial transferability.

Overall, these findings validate Assumption 1 that the truncation of gradients negatively impacts adversarial transferability. By preserving gradient information and carefully adjusting the replacement probabilities, it is possible to improve the effectiveness of adversarial attacks across different models.

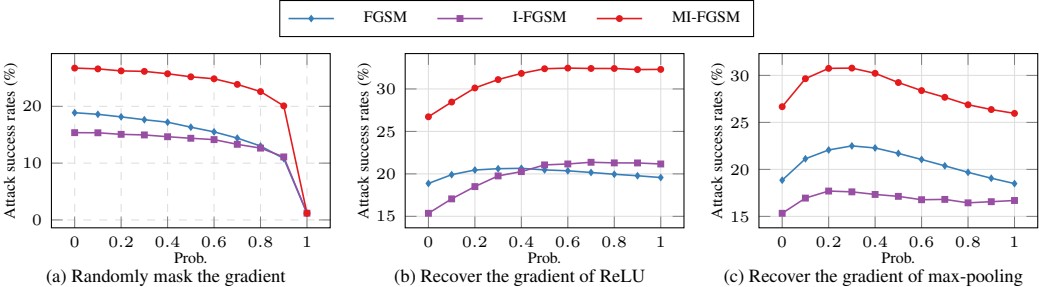

Figure 2: Average untargeted attack success rates (%) of FGSM, I-FGSM and MI-FGSM when we randomly mask the gradient, recover the gradient of ReLU or max-pooling layers, respectively. The adversarial examples are generated on ResNet-50 and tested on all the nine victim models illustrated in Sec. 4.1. Raw data is provided in Appendix A.1.

## 3.2 Mitigating the Negative Impact of Gradient Truncation

In Sec. 3.1, we demonstrate that reducing the probability of gradient truncation in non-linear layers can enhance adversarial transferability. However, setting all elements in the corresponding derivative to one is not optimal for generating transferable adversarial examples. Here we investigate how to modify the backward propagation process of non-linear layers to further enhance the transferability.

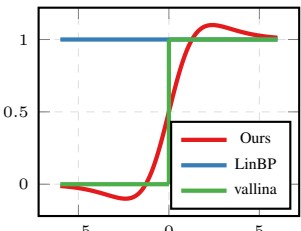

Figure 3: Various candidate derivatives of ReLU function.

Within the standard backward propagation procedure, the elements comprising the derivative depend on the magnitudes of the associated feature map. This observation provides an impetus for considering the intrinsic characteristics of the underlying features when diminishing the probability of gradient truncation. To this end, we modify the gradient calculation for the ReLU activation function and max-pooling in the backward propagation procedure as follows:

- **Gradient calculation for ReLU**. To ensure precise gradient calculation, it is important to exclude extreme values from consideration when calculating the gradient, while still maintaining the relationship between the elements in the derivative and the magnitude of the feature map. Among the family of ReLU activation functions, SiLU [14] provides a smooth and continuous gradient across the entire input range and is less susceptible to gradient saturation issues. Hence, we propose using the derivative of SiLU to calculate the gradient of ReLU during the backward propagation process, *i.e.*, $\frac{\partial z_{i+1}}{\partial z_i} = \sigma(z_i) \cdot (1 + z_i \cdot (1 - \sigma(z_i)))$, where $\sigma(\cdot)$ is the Sigmoid function. This formulation allows our gradient calculation to reflect the input magnitude mainly within the input range around $[-5, 5]$, while closely resembling the behavior of ReLU when the input is outside this range. As shown in Fig. 3, our proposed gradient calculation method demonstrates improved alignment with the input's magnitude compared to both the original derivative of ReLU and the derivative used in LinBP. By leveraging the smoothness and non-monotonicity of SiLU, we can obtain more accurate and reliable gradient information for ReLU.
- **Gradient calculation for max-pooling**. Similar to the gradient calculation for ReLU, it is essential to exclude extreme values and ensure that the gradient remains connected to the magnitude of the feature map. Furthermore, in the case of max-pooling, the summation of gradients within each window should remain at one to minimize modifications to the gradient. To address these considerations, we propose using the softmax function to calculate the gradient within each window $w$ of the max-pooling operation:

$$\left[ \frac{\partial z_{k+1}}{\partial z_k} \right]_{i,j,w} = \frac{e^{t \cdot z_{k,i,j}}}{\sum_{v \in w} e^{t \cdot v}}, \tag{3}$$

where $t$ is the temperature coefficient to adjust the smoothness of the gradient. If the feature $z_{k,i,j}$ is related to multiple windows (*i.e.*, the stride is smaller than the size of max-pooling), we sum its gradient calculated by Eq. 3 in each window as the final gradient.

| Attacker | Method | Inc-v3 | IncRes-v2 | DenseNet | MobileNet | PNASNet | SENet | Inc-v3$_{ens3}$ | Inc-v3$_{ens4}$ | IncRes-v2$_{ens}$ |
|---|---|---|---|---|---|---|---|---|---|---|
| PGD | N/A | 16.34 | 13.38 | 36.86 | 36.12 | 13.46 | 17.14 | 10.24 | 9.46 | 5.52 |
| | SGM | 23.68 | 19.82 | 51.66 | 55.44 | 22.12 | 30.34 | 13.78 | 12.38 | 7.90 |
| | LinBP | 27.22 | 23.04 | 59.34 | 59.74 | 22.68 | 33.72 | 16.24 | 13.58 | 7.88 |
| | Ghost | 17.74 | 13.68 | 42.36 | 41.06 | 13.92 | 19.10 | 11.60 | 10.34 | 6.04 |
| | **BPA** | **35.36** | **30.12** | **70.70** | **68.90** | **32.52** | **42.02** | **22.72** | **19.28** | **12.40** |
| MI-FGSM | N/A | 26.20 | 21.50 | 51.50 | 49.68 | 22.92 | 30.12 | 16.22 | 14.58 | 9.00 |
| | SGM | 33.78 | 28.84 | 63.06 | 65.84 | 31.90 | 41.54 | 19.56 | 17.48 | 10.98 |
| | LinBP | 35.92 | 29.82 | 68.66 | 69.72 | 30.24 | 41.68 | 19.98 | 16.58 | 9.94 |
| | Ghost | 29.76 | 23.68 | 57.28 | 56.10 | 25.00 | 34.76 | 17.10 | 14.76 | 9.50 |
| | **BPA** | **47.58** | **41.22** | **80.54** | **79.40** | **44.70** | **54.28** | **32.06** | **25.98** | **17.46** |
| VMI-FGSM | N/A | 42.68 | 36.86 | 68.82 | 66.68 | 40.78 | 46.34 | 27.36 | 24.20 | 17.18 |
| | SGM | 50.04 | 44.28 | 77.56 | 79.34 | 48.58 | 56.86 | 32.22 | 27.72 | 19.66 |
| | LinBP | 47.70 | 40.40 | 77.44 | 78.76 | 41.48 | 52.10 | 28.58 | 24.06 | 16.60 |
| | Ghost | 47.82 | 41.42 | 75.98 | 73.40 | 44.84 | 52.78 | 30.84 | 27.18 | 19.08 |
| | **BPA** | **55.00** | **48.72** | **85.44** | **83.64** | **52.02** | **60.88** | **38.76** | **33.70** | **23.78** |
| ILA | N/A | 29.10 | 26.08 | 58.02 | 59.10 | 27.60 | 39.16 | 15.12 | 12.30 | 7.86 |
| | SGM | 35.64 | 32.34 | 65.20 | 71.22 | 34.20 | 46.72 | 17.10 | 13.86 | 9.08 |
| | LinBP | 37.36 | 34.24 | 71.98 | 72.84 | 35.12 | 48.80 | 19.38 | 14.10 | 9.28 |
| | Ghost | 30.06 | 26.50 | 60.52 | 61.74 | 28.68 | 40.46 | 14.84 | 12.54 | 7.90 |
| | **BPA** | **47.62** | **43.50** | **81.74** | **80.88** | **47.88** | **60.64** | **27.94** | **20.64** | **14.76** |
| SSA | N/A | 35.78 | 29.58 | 60.46 | 64.70 | 25.66 | 34.18 | 20.64 | 17.30 | 11.44 |
| | SGM | 45.22 | 38.98 | 70.22 | 78.44 | 35.30 | 46.06 | 26.28 | 21.64 | 14.50 |
| | LinBP | 48.48 | 41.90 | 75.02 | 78.30 | 36.66 | 49.58 | 28.76 | 23.64 | 15.46 |
| | Ghost | 36.44 | 28.62 | 61.12 | 66.80 | 24.90 | 33.98 | 20.58 | 16.84 | 10.82 |
| | **BPA** | **51.36** | **44.70** | **76.24** | **79.66** | **39.38** | **50.00** | **32.10** | **26.44** | **18.20** |

Table 1: Untargeted attack success rates (%) of various adversarial attacks on nine models when generating the adversarial examples on ResNet-50 w/wo various model-related methods.

In practice, we adopt the above two strategies to calculate the gradient of ReLU and max-pooling during the backward propagation process. This approach allows us to circumvent the issue of gradient truncation introduced by these non-linear layers. We refer to this modified backward propagation technique as Backward Propagation Attack (BPA), which can be applied to existing CNNs to adapt to various transfer-based attack methods.

## 4 Experiments

In this section, we conduct extensive experiments on standard ImageNet dataset [35] to validate the effectiveness of the proposed BPA. We first specify our experimental setup, then we conduct a series of experiments to compare BPA with existing state-of-the-art attacks under different settings. Additionally, we provide ablation studies to further investigate the performance and behavior of BPA.

### 4.1 Experimental Setup

**Dataset.** Following LinBP [12], we randomly sample 5,000 images pertaining to the 1,000 categories from ILSVRC 2012 validation set [35], which could be classified correctly by all the victim models.

**Models.** We select ResNet-50 [13] and VGG-19 [37] as our surrogate model for generating adversarial examples. As for the victim models, we consider six standardly trained networks, *i.e.*, Inception-v3 (Inc-v3) [13], Inception-Resnet-v2 (IncRes-v2) [38], DenseNet [16], MobileNet-v2 [36], PNASNet [28], and SENet [15]. Additionally, we adopt three ensemble adversarially trained models, namely ens3-adv-Inception-v3 (Inc-v3$_{ens3}$), ens4-Inception-v3 (Inc-v3$_{ens4}$), and ens-adv-Inception-ResNet-v2 (IncRes-v2$_{ens}$) [40]. To address the issue of different input shapes required by these models, we adhere to the official pre-processing pipeline, including resizing and cropping techniques.

**Baselines.** We adopt three model-related methods as our baselines, *i.e.*, SGM [53], LinBP [12] and Ghost [23], and evaluate their performance to boost adversarial transferability of iterative attacks (PGD [31]), momentum-based attacks (MI-FGSM [6], VMI-FGSM [43]), advanced objective functions (ILA [17]) and input transformation-based attacks (SSA [30]).

**Hyper-parameters.** We adopt the maximum magnitude of perturbation $\epsilon = 8/255$ to align with existing works. We run the attacks in $T = 10$ iterations with step size $\alpha = 1.6/255$ for untargeted attacks and $T = 300$ iterations with step size $\alpha = 1/255$ for targeted attacks. We set the momentum decay factor $\mu = 1.0$ and sample 20 examples for VMI-FGSM. The number of spectrum transformations and tuning factor is set to $N = 20$ and $\rho = 0.5$, respectively. The decay factor for SGM is

| Attacker | Method | Inc-v3 | IncRes-v2 | DenseNet | MobileNet | PNASNet | SENet | Inc-v3$_{ens3}$ | Inc-v3$_{ens4}$ | IncRes-v2$_{ens}$ |
|---|---|---|---|---|---|---|---|---|---|---|
| | SGM | 23.68 | 19.82 | 51.66 | 55.44 | 22.12 | 30.34 | 13.78 | 12.38 | 7.90 |
| | SGM+BPA | **43.44** | **38.14** | **77.66** | **81.50** | **41.42** | **53.56** | **27.20** | **22.58** | **14.70** |
| | LinBP | 27.22 | 23.04 | 59.34 | 59.74 | 22.68 | 33.72 | 16.24 | 13.58 | 7.88 |
| PGD | LinBP+BPA | **39.08** | **34.80** | **77.80** | **76.86** | **40.50** | **50.26** | **25.66** | **22.46** | **15.10** |
| | Ghost | 17.74 | 13.68 | 42.36 | 41.06 | 13.92 | 19.10 | 11.60 | 10.34 | 6.04 |
| | Ghost+BPA | **34.62** | **29.28** | **69.48** | **69.20** | **29.98** | **41.60** | **22.68** | **18.88** | **11.48** |
| | SGM | 33.78 | 28.84 | 63.06 | 65.84 | 31.90 | 41.54 | 19.56 | 17.48 | 10.98 |
| | SGM+BPA | **56.04** | **49.10** | **85.32** | **88.08** | **52.96** | **63.30** | **36.10** | **29.78** | **20.98** |
| | LinBP | 35.92 | 29.82 | 68.66 | 69.72 | 30.24 | 41.68 | 19.98 | 16.58 | 9.94 |
| MI-FGSM | LinBP+BPA | **48.74** | **43.96** | **83.30** | **83.52** | **50.00** | **59.22** | **32.60** | **28.42** | **20.32** |
| | Ghost | 29.76 | 23.68 | 57.28 | 56.10 | 25.00 | 34.76 | 17.10 | 14.76 | 9.50 |
| | Ghost+BPA | **50.42** | **42.84** | **83.02** | **81.24** | **44.70** | **56.50** | **32.46** | **26.82** | **18.34** |

Table 2: Untargeted attack success rates (%) of various baselines combined with our method using PGD and MI-FGSM. The adversarial examples are generated on ResNet-50.

$\gamma = 0.5$ and the random range of Ghost network is $\lambda = 0.22$. We follow the setting of LinBP to modify the backward propagation of ReLU in the last eight residual blocks of ResNet-50. We set the temperature coefficient $t = 10$ for ResNet-50 and $t = 1$ for VGG-19.

## 4.2 Evaluation on Untargeted Attacks

To validate the effectiveness of our proposed method, we compare BPA with several other model-related methods (*i.e.*, SGM, LinBP, Ghost) on ResNet-50 to boost various adversarial attacks, namely PGD, MI-FGSM, VMI-FGSM, ILA and SSA. Here we adopt ResNet-50 as the surrogate model since SGM is specific to ResNets. However, it is worth noting that BPA is general to various surrogate models with non-linear layers and we also report the results on VGG-19 in Appendix A.2. To further validate the effectiveness of BPA, we also consider more input transformation based attacks, different perturbation budgets and conduct evaluations on CIFAR-10 dataset [20] in Appendix A.3-A.5. We measure the attack success rates by evaluating the misclassification rates of the nine different target models on the generated adversarial examples.

**Evaluations on the single baseline**. We can observe from Table 1 that the model-related strategies can consistently boost performance of the five typical attacks on nine models. Among the baseline methods, LinBP generally achieves the best performance, except for VMI-FGSM where SGM surpasses LinBP. By addressing the issue of gradient truncation, BPA consistently improves the performance of all the five attack methods and achieves the best overall performance. On average, BPA outperforms the runner-up attack by a significant margin of 7.84%, 11.19%, 5.08%, 9.17%, 2.25%, respectively. These results highlight the effectiveness and generality of BPA in generating transferable adversarial examples compared with existing model-related strategies. The performance improvement achieved by BPA on SGM and LinBP, which also modify the backward propagation, validates our hypothesis that reducing the gradient truncation introduced by non-linear layers is beneficial for enhancing the adversarial transferability. This emphasizes the importance of carefully considering the backward propagation procedure when generating transferable adversarial examples.

**Evaluations by combining BPA with the baselines**. The primary objective of BPA is to mitigate the negative impact of gradient truncation on adversarial transferability, which is not considered by the baselines. Hence, it is expected that BPA can also boost the performance of these baselines. For validation, we integrate BPA with the baseline methods to enhance the performance of PGD and MI-FGSM attacks. The results of these combinations are presented in Table 2. We can observe that BPA can effectively boost the adversarial transferability of various baselines. On average, BPA can boost the best baseline (*i.e.*, LinBP) with a remarkable margin of 13.23% and 20.94% for PGD and MI-FGSM, highlighting

| Attacker | Method | HGD | R&P | NIPS-r3 | JPEG | RS | NRP |
|---|---|---|---|---|---|---|---|
| | N/A | 9.34 | 5.00 | 6.00 | 11.04 | 8.50 | 11.96 |
| | SGM | 16.80 | 7.50 | 9.44 | 13.96 | 10.50 | 12.76 |
| PGD | LinBP | 16.80 | 7.68 | 10.08 | 15.76 | 10.50 | 13.14 |
| | Ghost | 9.60 | 5.06 | 6.42 | 11.92 | 9.50 | 12.06 |
| | BPA | **23.96** | **12.02** | **15.60** | **22.52** | **14.00** | **14.08** |
| | N/A | 16.64 | 8.04 | 9.92 | 16.68 | 13.00 | 13.32 |
| | SGM | 24.80 | 11.02 | 13.16 | 20.26 | 14.00 | 14.38 |
| MI-FGSM | LinBP | 21.98 | 10.32 | 13.26 | 20.56 | 12.50 | 13.22 |
| | Ghost | 17.98 | 8.88 | 10.64 | 18.52 | 13.50 | 13.84 |
| | BPA | **34.30** | **17.84** | **22.04** | **30.86** | **17.50** | **15.96** |

Table 3: Untargeted attack success rates (%) of several attacks on six defenses when generating the adversarial examples on ResNet-50 w/wo various model-related methods.

| Attacker | Method | Inc-v3 | IncRes-v2 | DenseNet | MobileNet | PNASNet | SENet | Inc-v3$_{ens3}$ | Inc-v3$_{ens4}$ | IncRes-v2$_{ens}$ |
|---|---|---|---|---|---|---|---|---|---|---|
| PGD | N/A | 0.54 | 0.80 | 4.48 | 2.04 | 1.62 | 2.26 | 0.18 | 0.08 | 0.02 |
| | SGM | 2.56 | 3.12 | 15.08 | 8.68 | 5.78 | 9.84 | 0.62 | 0.18 | 0.04 |
| | LinBP | 5.30 | 4.84 | 16.08 | 8.48 | 7.26 | 7.94 | 1.50 | 0.54 | 0.28 |
| | Ghost | 1.34 | 2.14 | 10.24 | 4.74 | 3.90 | 6.64 | 0.36 | 0.16 | 0.10 |
| | BPA | 8.76 | 9.74 | 23.76 | 13.42 | 14.66 | 13.76 | 2.52 | 1.02 | 0.72 |
| MI-FGSM | N/A | 0.16 | 0.26 | 2.06 | 0.90 | 0.42 | 1.22 | 0.00 | 0.02 | 0.02 |
| | SGM | 0.74 | 0.76 | 5.84 | 3.24 | 1.66 | 3.70 | 0.00 | 0.02 | 0.00 |
| | LinBP | 3.30 | 3.00 | 13.44 | 6.26 | 5.50 | 7.18 | 0.30 | 0.10 | 0.02 |
| | Ghost | 0.66 | 0.76 | 5.48 | 2.14 | 1.58 | 3.38 | 0.08 | 0.02 | 0.00 |
| | BPA | 5.68 | 7.30 | 23.34 | 12.16 | 12.50 | 14.56 | 0.60 | 0.12 | 0.06 |

Table 4: Targeted attack success rates (%) of various attackers on nine models when generating adversarial examples on ResNet-50 w/wo model-related methods using PGD and MI-FGSM.

the high effectiveness and superiority of BPA. Such high performance also validates its excellent generality to various architectures and supports our hypothesis about gradient truncation.

**Evaluations on defense methods**. To further evaluate the effectiveness of BPA, we also assess its performance on six defense methods using PGD and MI-FGSM, namely HGD [25], R&P [56], NIPS-r3[3], JPEG [11], RS [5] and NRP [32]. The results are presented in Table 3. We can observe that our BPA method successfully enhances both the PGD and MI-FGSM attacks, leading to higher attack performance against the defense methods. The results suggest that BPA can effectively enhance adversarial attacks against a range of defense techniques, reinforcing its potential as a powerful tool for generating transferable adversarial examples.

In summary, BPA exhibits superior transferability compared to various baseline methods when evaluated using a range of transfer-based attacks. It also exhibits good generality to further boost existing model-related approaches and achieves remarkable performance on several defense models, highlighting its effectiveness and versatility in generating highly transferable adversarial examples.

### 4.3 Evaluation on Targeted Attacks

To further evaluate the effectiveness of BPA, we also investigate its performance in boosting targeted attacks. Zhao *et al*. [60] identified that logit loss can yield better results than most resource-intensive attacks regarding targeted attacks. Here we adopt PGD and MI-FGSM to optimize the logit loss on ResNet-50 w/wo various model-related methods. The results are summarized in Table 4. Without the model-related methods, both PGD and MI-FGSM exhibit poor attack performance. However, when these methods are applied, the attack performance improves significantly. Notably, our BPA method achieves the best attack performance among all the baselines. This highlights the high effectiveness and excellent versatility of our proposed method in boosting targeted attacks and exhibits its potential to improve adversarial attacks in a wide range of scenarios. We also provide the results on VGG-19 in Appendix A.6.

### 4.4 Ablation Study

To gain further insights into the effectiveness of BPA, we perform parameter studies on two crucial aspects: the position of the first ReLU layer to be modified and the temperature coefficient $t$ for max-pooling. Additionally, we conduct ablation studies to investigate the impact of diminishing the gradient truncation of ReLU and max-pooling separately. We also provide more discussions about BPA in Appendix A.7-A.9.

**On the position of the first ReLU layer to be modified**. ReLU activation functions are densely applied in existing neural networks. For instance, there are totally 17 ReLU activation functions in ResNet-50. Intuitively, the truncation in the latter layers has a greater impact on gradient relevance compared to the earlier layers. As BPA aims to recover the truncated gradients by injecting imprecise gradients into the backward propagation, it is essential to focus on the more critical layers. To identify these important layers and evaluate their impact on transferability, we conduct the BPA attack using MI-FGSM by modifying the ReLU layers starting from the $i$-th layer, where $1 \le i \le 17$. As shown in Fig. 4a, modifying the last ReLU layer alone significantly improves the transferability of the attack, showing its high effectiveness. As we modify more ReLU layers, the transferability further

---
[3] https://github.com/anlthms/nips-2017/tree/master/mmd

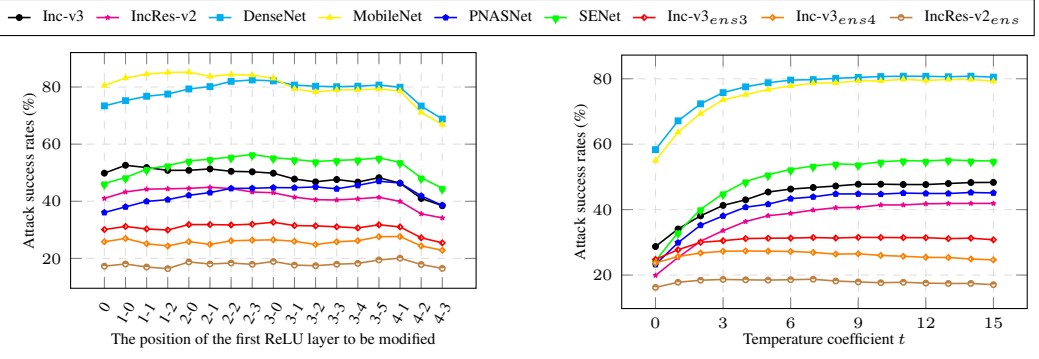

(a) Attack success rate (%) of BPA using MI-FGSM by modifying the ReLU layers starting from the $i$-th layer. Here 3-0 indicates the first ReLU layer in the third stage

(b) Attack success rate (%) of BPA using MI-FGSM with various temperature coefficients ($0 \leq t \leq 15$) in Eq. (3) for the max-pooling layer

Figure 4: Hyper-parameter studies on the position of the first ReLu layer to be modified and the temperature coefficient $t$ for the max-pooling layer.

| Attacker | ReLU | Max-pooling | Inc-v3 | IncRes-v2 | DenseNet | MobileNet | PNASNet | SENet | Inc-v3$_{ens3}$ | Inc-v3$_{ens4}$ | IncRes-v2$_{ens}$ |
|---|---|---|---|---|---|---|---|---|---|---|---|
| PGD | ✗ | ✗ | 16.34 | 13.38 | 36.86 | 36.12 | 13.46 | 17.40 | 10.24 | 9.46 | 5.52 |
|  | ✓ | ✗ | 29.38 | 24.00 | 62.80 | 61.82 | 24.98 | 34.96 | 17.52 | 14.38 | 8.90 |
|  | ✗ | ✓ | 20.26 | 16.16 | 44.66 | 42.82 | 17.12 | 21.52 | 13.20 | 11.88 | 7.74 |
|  | ✓ | ✓ | **35.36** | **30.12** | **70.70** | **68.90** | **32.52** | **42.02** | **22.72** | **19.28** | **12.40** |
| MI-FGSM | ✗ | ✗ | 26.20 | 21.50 | 51.50 | 49.68 | 22.92 | 30.12 | 16.22 | 14.58 | 9.00 |
|  | ✓ | ✗ | 41.50 | 34.42 | 74.96 | 74.42 | 35.96 | 47.58 | 23.34 | 18.22 | 10.94 |
|  | ✗ | ✓ | 34.16 | 29.02 | 61.38 | 59.42 | 32.24 | 37.32 | 21.74 | 19.96 | 14.70 |
|  | ✓ | ✓ | **47.58** | **41.22** | **80.54** | **79.40** | **44.70** | **54.28** | **32.06** | **25.98** | **17.46** |

Table 5: Untargeted attack success rates (%) of PGD and MI-FGSM when generating adversarial examples on ResNet-50 w/wo modifying the backward propagation of ReLU or max-pooling.

improves and remains consistently high for most models. However, for a few models (*e.g.*, PNASNet), modifying more ReLU layers leads to a slight decay on performance. To maintain a high level of performance across all nine models, we modify the ReLU layers starting from 3-0 ReLU layer, which is also adopted in LinBP [12].

**On the temperature coefficient $t$ for max-pooling**. The temperature coefficient $t$ plays a crucial role in determining the distribution of relative gradient magnitudes within each window. For example, when $t = 0$, the gradient distribution becomes a normalized uniform distribution. To find an appropriate temperature coefficient, we conduct the BPA attack using MI-FGSM with various temperatures. As shown in Fig. 4b, when $t = 0$, the attack exhibits the poorest performance but still outperforms the vanilla MI-FGSM. As we increase the value of $t$, the attack's performance consistently improves and reaches a high level of performance after $t = 10$. By selecting a suitable temperature coefficient, we ensure that the gradient distribution within each window is well-balanced and contributes effectively to the adversarial perturbation. Thus, we adopt $t = 10$ in our experiments.

**Ablation studies on ReLU and max-pooling**. As stated in Sec. 3.1, we hypothesize that the gradient truncation caused by non-linear layers, such as ReLU and max-pooling in ResNet-50, has a detrimental effect on adversarial transferability. To further validate this hypothesis, we conduct ablation studies by comparing the performance of PGD and MI-FGSM attacks using the vallina backward propagation, the backward propagation modified by either ReLU or max-pooling, and both modifications combined. As shown in Table 5, adopting the modified backward propagation with either ReLU or max-pooling results in a significant improvement in adversarial transferability for both PGD and MI-FGSM attacks. Considering the presence of only one max-pooling layer in ResNet-50, the average performance improvement of $4.07\%$ and $7.58\%$ for PGD and MI-FGSM highlights the high effectiveness of BPA and underscores the efficacy of BPA in addressing the issue of gradient truncation. Furthermore, when both ReLU and max-pooling layers are modified in backward propagation, PGD and MI-FGSM exhibit the best performance. This finding supports the rational design of BPA and highlights the importance of mitigating gradient truncation in both ReLU and max-pooling layers to achieve optimal adversarial transferability.

# 5   Conclusion

In this work, we analyzed the backward propagation procedure and identified that non-linear layers (*e.g.*, ReLU and max-pooling) introduce gradient truncation, which undermined the adversarial transferability. Based on this finding, we proposed a novel attack called Backward Propagation Attack (BPA) to mitigate the gradient truncation for more transferable adversarial examples. In particular, BPA addressed gradient truncation by introducing a non-monotonic function as the derivative of the ReLU activation function and incorporating softmax with temperature to calculate the derivative of max-pooling. These modifications helped to preserve the gradient information and prevented significant truncation during the backward propagation process. Empirical evaluations on ImageNet dataset demonstrated that BPA can significantly enhance existing untargeted and targeted attacks and outperformed the baselines by a remarkable margin. Our findings identified the vulnerability of model architectures and raised a new challenge in designing secure deep neural network architectures.

# 6   Limitation

Our proposed BPA modifies backpropagation process for gradient calculation, making it only suitable for gradient-based attacks. Besides, BPA modifies the derivatives of non-linear layers, such as ReLU and max-pooling. Consequently, it may not be directly applicable to models lacking these specific components, such as transformers. In the future, we will investigate how to generalize our BPA to such transformers by refining the derivatives of some components, *e.g.*, softmax. This endeavor to enhance the generality and versatility of BPA will be an essential aspect of ongoing research, paving the way for the broader applicability of the proposed method and facilitating its adoption in various deep learning models beyond those with ReLU and max-pooling layers.

# 7   Acknowledgement

This work is supported by National Natural Science Foundation (U22B2017, 62076105) and International Cooperation Foundation of Hubei Province, China (2021EHB011).

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

# A Appendix

In the appendix, we provide extra results about our proposed BPA. We first provide the raw data for Fig. 2. Considering that our main evaluations are conducted on ResNet-50 since the baseline SGM primarily focuses on the residual connection, we also conduct the evaluations on VGG-19 to further validate the effectiveness and generality of our proposed BPA. Then we provide the evaluations using more input transformation based attacks, and different perturbation budgets and also evaluate various methods on CIFAR-10 datasets. To further gain insights into the superiority of BPA, we also offer further discussions and explorations about the motivation and design.

## A.1 Raw Data for Fig. 2

Due to the limited space, we only report the average attack success rates in Fig. 2. To validate its generality to various deep models, we also report the raw data in Tab. A1-A3. We can see that the attacker exhibits the same trends on various deep models, which is consistent with the results in Fig. 2.

| Prob. | Inv-v3 | InvRes-v2 | DenseNet | MobileNet | PNASNet | SENet | Inc-v3$_{ens3}$ | Inc-v3$_{ens4}$ | IncRes-v2$_{ens}$ |
|---|---|---|---|---|---|---|---|---|---|
| 0.00 | 26.38 | 21.38 | 51.26 | 49.62 | 22.64 | 30.22 | 15.94 | 14.44 | 8.68 |
| 0.10 | 26.14 | 21.62 | 51.30 | 49.78 | 22.12 | 29.32 | 15.92 | 14.24 | 8.88 |
| 0.20 | 25.72 | 21.56 | 50.72 | 49.02 | 22.18 | 28.90 | 15.82 | 13.82 | 8.38 |
| 0.30 | 25.62 | 21.24 | 50.94 | 48.74 | 21.84 | 29.12 | 15.52 | 13.82 | 8.54 |
| 0.40 | 24.96 | 20.78 | 50.02 | 48.32 | 21.48 | 28.64 | 15.04 | 13.92 | 8.46 |
| 0.50 | 24.94 | 20.16 | 49.52 | 47.66 | 20.64 | 27.92 | 14.42 | 13.22 | 8.28 |
| 0.60 | 24.94 | 19.70 | 48.28 | 47.20 | 20.34 | 27.92 | 14.28 | 12.84 | 8.14 |
| 0.70 | 23.18 | 18.78 | 46.96 | 46.08 | 18.92 | 26.42 | 13.96 | 12.72 | 7.74 |
| 0.80 | 22.84 | 17.18 | 44.52 | 44.38 | 17.52 | 24.92 | 13.28 | 11.72 | 7.00 |
| 0.90 | 20.22 | 15.78 | 39.82 | 41.16 | 14.80 | 20.84 | 11.22 | 10.54 | 6.34 |
| 1.00 | 0.02 | 1.10 | 0.00 | 0.02 | 0.02 | 0.02 | 3.76 | 4.30 | 1.28 |

Table A1: Original data for Fig. 2a

| Prob. | Inv-v3 | InvRes-v2 | DenseNet | MobileNet | PNASNet | SENet | Inc-v3$_{ens3}$ | Inc-v3$_{ens4}$ | IncRes-v2$_{ens}$ |
|---|---|---|---|---|---|---|---|---|---|
| 0.00 | 26.38 | 21.38 | 51.26 | 49.62 | 22.64 | 30.22 | 15.90 | 14.42 | 8.68 |
| 0.10 | 28.12 | 23.22 | 54.40 | 51.62 | 24.82 | 32.38 | 16.98 | 15.34 | 9.26 |
| 0.20 | 29.34 | 24.80 | 57.28 | 55.02 | 26.60 | 34.14 | 17.70 | 16.12 | 10.04 |
| 0.30 | 30.32 | 25.70 | 59.30 | 56.52 | 27.48 | 36.00 | 17.92 | 16.26 | 10.42 |
| 0.40 | 31.34 | 26.36 | 60.50 | 57.78 | 28.66 | 36.68 | 18.74 | 16.24 | 10.18 |
| 0.50 | 31.86 | 26.76 | 60.74 | 58.64 | 29.46 | 37.68 | 19.12 | 16.74 | 10.42 |
| 0.60 | 31.68 | 27.02 | 61.00 | 58.66 | 29.80 | 37.58 | 18.94 | 16.66 | 10.76 |
| 0.70 | 31.90 | 26.36 | 60.62 | 59.40 | 29.52 | 37.90 | 18.88 | 16.72 | 10.40 |
| 0.80 | 31.42 | 26.82 | 61.16 | 59.16 | 29.42 | 37.90 | 18.80 | 16.40 | 10.60 |
| 0.90 | 31.80 | 26.56 | 60.42 | 58.90 | 29.06 | 37.50 | 18.98 | 16.78 | 10.52 |
| 1.00 | 32.00 | 26.52 | 60.44 | 59.44 | 29.28 | 37.46 | 18.78 | 16.52 | 10.30 |

Table A2: Raw data for Fig. 2b

| Prob. | Inv-v3 | InvRes-v2 | DenseNet | MobileNet | PNASNet | SENet | Inc-v3$_{ens3}$ | Inc-v3$_{ens4}$ | IncRes-v2$_{ens}$ |
|---|---|---|---|---|---|---|---|---|---|
| 0.00 | 26.08 | 21.40 | 51.30 | 49.64 | 22.50 | 30.26 | 15.74 | 14.44 | 8.68 |
| 0.10 | 28.64 | 23.78 | 56.14 | 53.10 | 26.28 | 32.78 | 18.20 | 16.64 | 11.30 |
| 0.20 | 29.66 | 24.14 | 56.98 | 54.72 | 27.18 | 33.54 | 19.62 | 18.04 | 12.86 |
| 0.30 | 28.92 | 24.06 | 57.16 | 54.36 | 27.74 | 31.58 | 20.50 | 19.28 | 13.40 |
| 0.40 | 28.22 | 23.46 | 56.12 | 53.16 | 26.72 | 30.24 | 20.62 | 19.44 | 14.04 |
| 0.50 | 27.50 | 21.68 | 54.40 | 52.06 | 25.54 | 28.48 | 20.34 | 19.02 | 14.14 |
| 0.60 | 26.70 | 20.72 | 53.02 | 50.10 | 24.24 | 27.42 | 19.92 | 19.30 | 14.04 |
| 0.70 | 25.82 | 20.14 | 51.38 | 48.36 | 22.94 | 26.12 | 20.24 | 19.70 | 14.34 |
| 0.80 | 25.00 | 19.04 | 50.02 | 47.36 | 21.92 | 24.12 | 20.22 | 19.68 | 14.56 |
| 0.90 | 24.70 | 18.34 | 48.98 | 46.36 | 21.36 | 23.34 | 20.38 | 19.68 | 14.18 |
| 1.00 | 24.04 | 18.10 | 48.34 | 45.08 | 21.02 | 22.40 | 20.38 | 20.10 | 14.18 |

Table A3: Raw data for Fig. 2c

| Attacker | Method | Inc-v3 | IncRes-v2 | DenseNet | MobileNet | PNASNet | SENet | Inc-v3$_{ens3}$ | Inc-v3$_{ens4}$ | IncRes-v2$_{ens}$ |
|---|---|---|---|---|---|---|---|---|---|---|
| PGD | N/A | 12.52 | 9.70 | 25.82 | 32.20 | 13.18 | 13.82 | 7.64 | 7.60 | 4.14 |
|  | LinBP | 13.52 | 10.28 | 27.60 | 34.36 | 14.16 | 15.12 | 8.32 | 7.88 | 4.20 |
|  | Ghost | 13.18 | 9.72 | 25.78 | 32.50 | 12.80 | 13.68 | 8.12 | 7.90 | 4.48 |
|  | **BPA** | **26.24** | **27.06** | **47.98** | **58.22** | **34.08** | **31.42** | **15.52** | **14.06** | **8.78** |
| MI-FGSM | N/A | 19.74 | 15.32 | 37.02 | 43.42 | 21.16 | 23.02 | 11.46 | 10.08 | 5.96 |
|  | LinBP | 20.28 | 15.24 | 36.84 | 44.44 | 20.66 | 23.28 | 10.92 | 9.52 | 5.48 |
|  | Ghost | 19.88 | 15.34 | 36.44 | 43.20 | 21.84 | 24.06 | 11.54 | 10.30 | 6.00 |
|  | **BPA** | **36.88** | **29.98** | **61.10** | **68.58** | **45.98** | **43.06** | **21.44** | **17.68** | **11.94** |
| VMI-FGSM | N/A | 37.20 | 29.58 | 58.20 | 62.20 | 40.88 | 38.86 | 21.14 | 17.62 | 11.10 |
|  | LinBP | 36.18 | 28.86 | 55.40 | 62.46 | 38.38 | 39.14 | 19.20 | 17.18 | 10.92 |
|  | Ghost | 36.94 | 29.75 | 58.32 | 62.16 | 41.32 | 38.96 | 21.18 | 17.58 | 11.20 |
|  | **BPA** | **51.60** | **43.00** | **74.08** | **78.74** | **59.54** | **54.74** | **32.88** | **30.04** | **20.18** |
| ILA | N/A | 16.08 | 13.8 | 31.28 | 42.62 | 19.72 | 25.16 | 8.76 | 7.70 | 4.62 |
|  | LinBP | 17.08 | 14.54 | 32.74 | 44.40 | 20.16 | 27.08 | 8.44 | 7.92 | 4.54 |
|  | Ghost | 16.56 | 14.08 | 31.80 | 41.90 | 20.12 | 25.98 | 8.84 | 7.84 | 4.76 |
|  | **BPA** | **29.70** | **25.06** | **50.84** | **61.52** | **38.84** | **41.20** | **15.30** | **12.36** | **8.30** |
| SSA | N/A | 33.52 | 26.38 | 50.86 | 60.26 | 30.94 | 30.78 | 17.06 | 14.52 | 8.78 |
|  | LinBP | 35.70 | 28.08 | 53.76 | 63.52 | 32.32 | 34.18 | 18.64 | 16.10 | 9.36 |
|  | Ghost | 33.52 | 25.92 | 51.31 | 60.50 | 30.96 | 30.02 | 17.16 | 14.74 | 8.74 |
|  | **BPA** | **50.16** | **40.68** | **70.90** | **78.86** | **51.64** | **47.86** | **29.52** | **26.50** | **18.30** |

Table A4: Untargeted attack success rates (%) of various adversarial attacks on nine models when generating the adversarial examples on VGG-19 w/wo various model-related methods.

| Attacker | Method | Inc-v3 | IncRes-v2 | DenseNet | MobileNet | PNASNet | SENet | Inc-v3$_{ens3}$ | Inc-v3$_{ens4}$ | IncRes-v2$_{ens}$ |
|---|---|---|---|---|---|---|---|---|---|---|
| PGD | LinBP | 13.52 | 10.28 | 27.60 | 34.36 | 14.16 | 15.12 | 8.32 | 7.88 | 4.20 |
|  | **LinBP+BPA** | **21.10** | **16.48** | **40.92** | **50.54** | **25.28** | **24.86** | **12.34** | **11.86** | **7.16** |
|  | Ghost | 13.18 | 9.72 | 25.78 | 32.50 | 12.80 | 13.68 | 8.12 | 7.90 | 4.48 |
|  | **Ghost+BPA** | **26.34** | **20.22** | **49.14** | **58.02** | **34.96** | **31.22** | **15.60** | **13.60** | **8.56** |
| MI-FGSM | LinBP | 20.28 | 15.24 | 36.84 | 44.44 | 20.66 | 23.28 | 10.92 | 9.52 | 5.48 |
|  | **LinBP+BPA** | **32.08** | **24.58** | **53.24** | **63.16** | **36.52** | **36.82** | **17.18** | **15.60** | **10.22** |
|  | Ghost | 19.88 | 15.34 | 36.44 | 43.20 | 21.84 | 24.06 | 11.54 | 10.30 | 6.00 |
|  | **Ghost+BPA** | **37.12** | **30.50** | **60.60** | **69.00** | **45.80** | **43.10** | **21.28** | **17.38** | **11.92** |

Table A5: Untargeted attack success rates (%) of various baselines combined with our method using PGD and MI-FGSM. The adversarial examples are generated on VGG-19.

## A.2 Additional Evaluation on Untargeted Attacks

To validate the generality of BPA to various architectures, we further validate the effectiveness of our proposed BPA on VGG-19. Specifically, we first conduct untargeted attacks on VGG-19 following the setting in Sec. 4.2. Here we take LinBP and Ghost as our baselines.

**Evaluations on the single baseline**. As shown in Table A4, model-related methods consistently achieve better attack performance than the attacks on the original models, showing the effectiveness of these methods. Compared with LinBP and Ghost, our proposed BPA exhibits superior performance across all five attacks. On average, BPA outperforms the runner-up method with a remarkable margin of 14.21%, 16.44%, 14.15%, 11.80%, 13.64% for PGD, MI-FGSM, VMI-FGSM, ILA, and SSA, respectively. These results are consistent with the findings reported in Sec. 4.2 for ResNet-50. The superior performance of BPA not only validates its effectiveness but also highlights its generality to different architectures.

**Evaluations by combining BPA with the baselines**. Similar in Sec. 4.2, we also integrate BPA into LinBP and Ghost to further boost the performance. The results in Table A5 indicate that BPA can significantly improve the attack performance of PGD and MI-FGSM. For instance, considering MI-FGSM attack, integrating BPA results in a clear performance improvement of 11.42% and 16.46% for LinBP and Ghost, respectively. These findings are consistent with the results obtained on ResNet-50, as discussed in Sec. 4.2. These results further highlight the effectiveness and superiority of BPA in boosting the adversarial transferability of existing attacks, which are not limited to the surrogate models.

**Evaluations on defense methods**. Finally, we evaluate these model-related approaches on defense methods and report the results in Table A6. Notably, our BPA method consistently enhances the performance of PGD and MI-FGSM attacks, yielding superior results against the defense methods compared to other model-related methods. On average, BPA outperforms the runner-up method with a

| Attacker | Method | Inc-v3 | IncRes-v2 | DenseNet | MobileNet | PNASNet | SENet | Inc-v3$_{ens3}$ | Inc-v3$_{ens4}$ | IncRes-v2$_{ens}$ |
|---|---|---|---|---|---|---|---|---|---|---|
| DIM | N/A | 45.00 | 38.56 | 71.64 | 70.08 | 41.60 | 48.56 | 28.52 | 24.82 | 16.48 |
| | SGM | 52.72 | 45.22 | 79.42 | 82.34 | 49.50 | 58.66 | 32.42 | 28.20 | 19.26 |
| | LinBP | 45.74 | 38.38 | 76.34 | 77.58 | 39.28 | 50.20 | 27.40 | 22.60 | 15.50 |
| | Ghost | 45.20 | 37.86 | 72.70 | 72.34 | 40.32 | 48.50 | 27.44 | 23.98 | 15.78 |
| | **BPA** | **59.20** | **50.86** | **87.70** | **86.92** | **55.40** | **62.68** | **40.32** | **34.84** | **24.42** |
| TIM | N/A | 32.58 | 26.66 | 58.44 | 55.44 | 29.26 | 34.84 | 21.38 | 18.76 | 13.54 |
| | SGM | 41.18 | 35.22 | 71.12 | 72.20 | 41.24 | 47.88 | 25.50 | 23.66 | 16.28 |
| | LinBP | 43.20 | 36.30 | 75.08 | 74.08 | 39.14 | 46.90 | 27.62 | 23.62 | 16.74 |
| | Ghost | 37.08 | 29.36 | 66.48 | 63.26 | 33.26 | 39.74 | 23.58 | 21.14 | 14.54 |
| | **BPA** | **59.20** | **50.86** | **87.70** | **86.92** | **55.40** | **62.68** | **40.32** | **34.84** | **24.42** |
| SIN | N/A | 42.68 | 33.76 | 70.12 | 65.70 | 34.90 | 39.90 | 26.36 | 23.48 | 15.22 |
| | SGM | 52.78 | 43.04 | 79.92 | 80.22 | 45.10 | 52.70 | 31.94 | 27.44 | 18.74 |
| | LinBP | 50.46 | 41.06 | 78.22 | 77.14 | 38.82 | 47.64 | 30.38 | 25.08 | 17.14 |
| | Ghost | 47.46 | 37.92 | 77.14 | 72.82 | 38.66 | 46.02 | 29.14 | 24.58 | 16.28 |
| | BPA | 52.40 | 43.44 | 80.12 | 76.38 | 45.42 | 50.28 | 39.16 | 36.32 | 26.06 |
| | **SGM+BPA** | **62.26** | **54.12** | **88.04** | **88.28** | **56.80** | **63.46** | **46.08** | **39.80** | **30.68** |
| DA | N/A | 45.00 | 38.56 | 71.64 | 70.08 | 41.60 | 48.56 | 28.52 | 24.82 | 16.48 |
| | SGM | 52.72 | 45.22 | 79.42 | 82.34 | 49.50 | 58.66 | 32.42 | 28.20 | 19.26 |
| | LinBP | 45.74 | 38.38 | 76.34 | 77.58 | 39.28 | 50.20 | 27.40 | 22.60 | 15.50 |
| | Ghost | 45.20 | 37.86 | 72.70 | 72.34 | 40.32 | 48.50 | 27.44 | 23.98 | 15.78 |
| | **BPA** | **59.20** | **50.86** | **87.70** | **86.92** | **55.40** | **62.68** | **40.32** | **34.84** | **24.42** |

Table A7: Untargeted attack success rates (%) of various input-transformation-based attacks on nine models when generating the adversarial examples on ResNet-50 w/wo various model-related methods.

margin of 5.32% and 7.78% for PGD and MI-FGSM, respectively. These findings further underscore the high effectiveness of BPA in improving the performance of various attacks and highlight its versatility in enhancing adversarial attacks across different architectural models.

In conclusion, the results obtained for untargeted attacks on VGG-19 align with the findings presented for ResNet-50 in Sec. 4.2. The significant and consistent improvement in performance across various architectures validates our motivation that addressing the gradient truncation issue caused by non-linear layers can enhance adversarial transferability. These findings also strongly support the high effectiveness and utility of our BPA to boost adversarial transferability.

| Attacker | Method | HGD | R&P | NIPS-r3 | JPEG | RS | NRP |
|---|---|---|---|---|---|---|---|
| PGD | N/A | 5.44 | 3.16 | 3.54 | 8.36 | 8.45 | 11.26 |
| | LinBP | 5.28 | 3.26 | 3.88 | 9.14 | 9.00 | 11.76 |
| | Ghost | 5.68 | 3.16 | 3.70 | 9.10 | 8.50 | 10.98 |
| | **BPA** | **15.78** | **7.58** | **9.46** | **16.22** | **12.00** | **13.18** |
| MI-FGSM | N/A | 9.12 | 5.08 | 5.76 | 12.18 | 8.00 | 12.86 |
| | LinBP | 8.06 | 4.75 | 5.34 | 11.56 | 8.50 | 12.32 |
| | Ghost | 9.14 | 4.92 | 5.78 | 12.32 | 8.50 | 12.08 |
| | **BPA** | **24.36** | **11.50** | **14.30** | **22.38** | **14.00** | **13.12** |

Table A6: Untargeted attack success rates (%) of several attacks on six defenses when generating the adversarial examples on VGG-19 w/wo various model-related methods.

## A.3 Additional Evaluation on Various Input Transformation based Attacks

In Table 1, we compare our BPA with various model-related attacks when combined with other attacks, including gradient-based (PGD), momentum-based (MI-FGSM, VMI-FGSM), objective-related (ILA) and input transformation based (SSA) attack. Due to the page limit, we only evaluate BPA with one up-to-date transformation based attack (SSA). Here we further evaluate its generality to other input transformation based attacks, namely DIM [57], TIM [7], SIN [26] and DA [18] in Table A7. As we can see, our BPA can significantly boost these input transformation based attacks. Compared with existing model-related attacks, BPA consistently exhibits better transferability for TIM, DIM, and DA. For SIN, BPA exhibits comparable attack performance with SGM on standardly trained models but much better transferability on adversarially trained models. To integrate the advantage of SGM and our BPA, we combine BPA with SGM for SIN, which outperforms the baselines with a clear margin. These results further validate its superiority in boosting adversarial transferability.

## A.4 Additional Evaluation on Another Widely Adopted Perturbation Budget

Both $8/255$ and $16/255$ are widely adopted perturbation budgets in adversarial learning. In Sec. 4.2, we mainly adopt $\epsilon = 8/255$ for evaluation. To further validate the effectiveness of our BPA, we also

evaluate BPA using PGD with $\epsilon = 16/255$. As shown in Table A8, BPA consistently outperforms the baselines with a clear margin, showing its remarkable effectiveness in boosting adversarial transferability with various perturbation budgets.

| Attacker | Method | Inc-v3 | IncRes-v2 | DenseNet | MobileNet | PNASNet | SENet | Inc-v3$_{ens3}$ | Inc-v3$_{ens4}$ | IncRes-v2$_{ens}$ |
|---|---|---|---|---|---|---|---|---|---|---|
| PGD | N/A | 36.42 | 29.90 | 65.08 | 62.46 | 30.40 | 38.60 | 21.26 | 18.70 | 12.60 |
| | SGM | 49.62 | 41.60 | 77.78 | 80.14 | 46.04 | 56.24 | 29.28 | 24.18 | 17.20 |
| | LinBP | 60.48 | 53.12 | 87.24 | 86.60 | 53.84 | 67.74 | 36.60 | 28.86 | 20.24 |
| | Ghost | 39.90 | 31.42 | 67.98 | 67.84 | 32.26 | 40.74 | 22.62 | 19.68 | 13.76 |
| | BPA | **72.06** | **66.56** | **93.42** | **92.28** | **70.26** | **78.74** | **49.76** | **40.80** | **31.08** |

Table A8: Untargeted attack success rates (%) of PGD on nine models when generating adversarial examples on ResNet-50 w/wo model-related methods using $\epsilon = 16/255$.

## A.5 Additional Evaluation on CIFAR-10 Dataset

Existing transfer-based attacks mainly validate their effectiveness on ImageNet dataset. For a fair comparison, we also evaluate our BPA on ImageNet dataset. Here we also conduct experiments on CIFAR-10 using PGD with $\epsilon = \frac{8}{255}$ on VGG-19. As shown in Table A9, BPA exhibits better transferability than the baselines, showing its high effectiveness and generality to various datasets and models.

| Attacker | Method | WRN | ResNeXt | DenseNet | pyramidnet | gdas |
|---|---|---|---|---|---|---|
| PGD | N/A | 65.94 | 65.66 | 63.56 | 16.96 | 50.00 |
| | LinBP | 67.88 | 67.46 | 65.02 | 18.18 | 51.78 |
| | Ghost | 66.46 | 65.52 | 62.92 | 17.32 | 49.18 |
| | BPA | **74.38** | **73.80** | **69.66** | **20.74** | **57.16** |

Table A9: Untargeted attack success rates (%) of several attacks on five models for CIFAR-10 dataset when generating the adversarial examples on VGG-19 w/wo various model-related methods.

## A.6 Additional Evaluation on Targeted Attacks

Targeted attacks are more challenging than untargeted attacks. To further validate the effectiveness and generality of BPA, we also perform the targeted attack on VGG-19, following the experimental settings in Sec. 4.3. The results are summarized in Table A10. It is interesting that LinBP decays the targeted attack performance on VGG-19. Since there is no skip connection in VGG-19, LinBP only modifies the derivative of ReLU, which might introduce an imprecise gradient. This highlights the significance that BPA excludes extreme values from consideration when calculating the gradient for better transferability. It is evident that our BPA achieves the best attack performance among various methods. Overall, BPA outperforms LinBP and Ghost by $8.18\%$ and $7.90\%$ for PGD, and $2.43\%$ and $2.46\%$ for MI-FGSM. These results further validate the effectiveness of BPA in targeted attacks, demonstrating its superiority over the baselines. The improved performance of BPA showcases its potential and generality in enhancing targeted attacks on various models.

| Attacker | Method | Inc-v3 | IncRes-v2 | DenseNet | MobileNet | PNASNet | SENet | Inc-v3$_{ens3}$ | Inc-v3$_{ens4}$ | IncRes-v2$_{ens}$ |
|---|---|---|---|---|---|---|---|---|---|---|
| PGD | N/A | 1.26 | 1.26 | 3.80 | 2.72 | 5.10 | 4.32 | 0.10 | 0.04 | 0.02 |
| | LinBP | 1.26 | 1.22 | 3.44 | 2.24 | 4.26 | 3.52 | 0.26 | 0.12 | 0.02 |
| | Ghost | 1.34 | 1.26 | 3.88 | 2.52 | 5.26 | 4.40 | 0.12 | 0.06 | 0.02 |
| | BPA | **6.70** | **7.30** | **19.44** | **12.56** | **23.34** | **17.32** | **1.80** | **0.76** | **0.74** |
| MI-FGSM | N/A | 0.18 | 0.10 | 1.00 | 0.92 | 1.02 | 1.14 | 0.00 | 0.00 | 0.02 |
| | LinBP | 0.24 | 0.20 | 1.18 | 0.86 | 0.94 | 1.02 | 0.02 | 0.00 | 0.02 |
| | Ghost | 0.22 | 0.14 | 0.94 | 0.74 | 1.04 | 1.12 | 0.00 | 0.02 | 0.02 |
| | BPA | **1.24** | **1.24** | **5.60** | **4.22** | **7.06** | **6.80** | **0.12** | **0.02** | **0.04** |

Table A10: Targeted attack success rates (%) of various attackers on nine models when generating adversarial examples on VGG-19 w/wo model-related methods using PGD and MI-FGSM.

## A.7 Additional Evaluation on Modifications of ReLU Layers with BPA and LinBP

In this work, we find that the truncation of non-linear layers (e.g., ReLU, max-pooling) decays the relevance between the gradient *w.r.t.* the input and loss. Making the model more linear can enhance such relevance, thus improving the transferability. However, making the model more linear is not optimal. Taking ReLU for example, we compare the transferability of PGD on LinBP and BPA by

solely changing the derivatives of ReLU. Here LinBP makes the model more linear than our BPA. As shown in Table A11, BPA exhibits better transferability than LinBP, which supports our argument.

| Attacker | Method | Inc-v3 | IncRes-v2 | DenseNet | MobileNet | PNASNet | SENet | Inc-v3$_{ens3}$ | Inc-v3$_{ens4}$ | IncRes-v2$_{ens}$ |
|----------|--------|--------|-----------|----------|-----------|---------|-------|-----------------|-----------------|-------------------|
| PGD | N/A | 16.34 | 13.38 | 36.86 | 36.12 | 13.46 | 17.14 | 10.24 | 9.46 | 5.52 |
| | LinBP | 27.22 | 23.04 | 59.34 | 59.74 | 22.68 | 33.72 | 16.24 | 13.58 | 7.88 |
| | **BPA** | **29.38** | **24.00** | **62.80** | **61.82** | **24.98** | **34.96** | **17.52** | **14.38** | **8.90** |

Table A11: Untargeted attack success rates (%) of PGD on nine models when generating adversarial examples on ResNet-50 with modifications on ReLU layers.

## A.8 Comparison between Random Replacement and BPA on Max-pooling Layers

For max-pooling, the initial replacement of zeros in the gradient with ones results in an increase in relevance and subsequent improvement in attack performance. However, as the number of replaced zeros increases, the vallina BP faces challenges in accurately discerning which elements are critical that are related to the magnitude of input values. This indiscriminate replacement introduces a substantial error, leading to a decay in attack performance. Hence, we also compare randomly replacing the zeros with ones using the probability of 0.3 (the best reult in Fig. 2c) and BPA for max-pooling on ResNet-50. As shown in Table A12, BPA exhibits better transferability, which further validates our motivation and indicates that the effectiveness of our BPA is not solely attributed to achieving linearity.

| Attacker | Method | Inc-v3 | IncRes-v2 | DenseNet | MobileNet | PNASNet | SENet | Inc-v3$_{ens3}$ | Inc-v3$_{ens4}$ | IncRes-v2$_{ens}$ |
|----------|--------|--------|-----------|----------|-----------|---------|-------|-----------------|-----------------|-------------------|
| PGD | N/A | 16.34 | 13.38 | 36.86 | 36.12 | 13.46 | 17.14 | 10.24 | 9.46 | 5.52 |
| | Replace(0.3) | 14.52 | 11.94 | 37.52 | 36.02 | 13.84 | 17.28 | 10.34 | 10.56 | 6.48 |
| | **BPA** | **20.26** | **16.16** | **44.66** | **42.82** | **17.12** | **21.52** | **13.20** | **11.88** | **7.74** |

Table A12: Untargeted attack success rates (%) of PGD on nine models when generating adversarial examples on ResNet-50 with modifications on max-pooling layers.

## A.9 Relevance between Gradient *w.r.t.* Input and Loss Function

The relevance between gradient *w.r.t.* input and loss function indicates the sensitivity of the loss function to changes in the input when taking a small step in the direction of gradient. This relevance can be formally defined as follows:

**Definition 1 (Relevance between gradient *w.r.t.* input and loss function)** *Given an input $x$, loss function $J(x)$ and a step size $\epsilon$, the relevance between gradient* w.r.t. *input and objective function can be defined as $\frac{J(x+\epsilon \cdot \nabla_x J(x)) - J(x)}{\epsilon}$.*

ReLU helps address the vanishing gradient issue during the training of deep models by eliminating some gradients. However, BPA focuses on boosting transferability of adversarial examples generated on such ReLU-activated networks. To effectively generate adversarial examples, it is crucial that the gradient *w.r.t.* the input provides a reliable direction to maximize the loss. Unfortunately, the truncation of ReLU makes the calculated gradient unable to provide such exactly precise direction, *e.g.*, making the gradient weakened to some extent. Similarly, max-pooling also truncates the gradient during the backpropagation, causing the gradient unable to indicate an exactly precise direction.

As summarized above, the truncation of ReLU and max-pooling drops gradient (introduces zeros) in the backpropagation process, which decays the relevance. We also calculate the Relevance using vallina backpropagation and our backpropagation on ResNet-50 using $1,000$ images. BPA achieved the relevance of $240.51$, while the gradient calculated by the vallina backpropgation achieves lower relevance($149.23$), which supports our hypothesis that non-linear layers can *decay the relevance*.

