# OpenReview forum: "Rethinking the Backward Propagation for Adversarial Transferability"
_NeurIPS.cc/2023/Conference — NeurIPS 2023 poster_

### Official Review · Reviewer_YaZN · 2023-06-22

**Soundness:** 2 fair
**Presentation:** 3 good
**Contribution:** 2 fair
**Rating:** 6
**Confidence:** 4

**Summary:**

This paper presents the Backward Propagation Attack (BPA) as a solution to the issue of gradient truncation in transfer-based attacks within adversarial machine learning. Through empirical validation, the authors highlight the negative impact of gradient truncation on adversarial transferability and propose modifications to the backward propagation process. BPA effectively enhances the relevance of the gradient between the loss function and the input, resulting in significant improvements in various untargeted and targeted transfer-based attacks compared to baseline approaches. This work not only provides valuable insights for enhancing adversarial transferability but also suggests promising directions for bolstering model robustness.

**Strengths:**

The paper demonstrates originality by identifying and addressing the issue of gradient truncation in transfer-based attacks, which has not been extensively explored before. The proposed Backward Propagation Attack (BPA) introduces modifications to the backward propagation process, mitigating the detrimental effect of gradient truncation. The empirical evaluation on the widely used ImageNet dataset showcases the high quality of the research, with rigorous experimentation and comparisons against baselines. The clarity of the paper is commendable, effectively communicating the problem, the proposed solution, and the experimental methodology. The findings are significant as they contribute to advancing transfer-based attacks, provide new insights into adversarial transferability, and have potential implications for improving model robustness.

**Weaknesses:**

While the paper presents valuable contributions, there are a few areas that could be improved. Firstly, the theoretical underpinnings of the proposed modifications to the gradient truncation during the backward propagation process could be further elaborated to enhance the understanding of the approach. Additionally, the limitations of the proposed techniques could be further discussed to provide a comprehensive view of its effectiveness, such as the potential performance decrease caused by replacing the ReLU function. Furthermore, the paper focuses mainly on the ImageNet dataset, and it would be beneficial to explore the generalizability of the approach on other datasets or domains. Overall, addressing these weaknesses would strengthen the paper and provide further insights into the proposed approach.

**Questions:**

1.	The paper proposes modifications to the backward propagation process to mitigate gradient truncation. Could the authors provide more theoretical justification for the choice of these specific modifications?
2.	The experimental evaluation is conducted on the ImageNet dataset. Have you considered testing the proposed BPA method on other datasets to assess its generalizability? It would be valuable to understand how the approach performs across different domains and datasets.
3.	It would be beneficial to discuss the limitations of the proposed BPA method. Are there any specific scenarios or conditions where the effectiveness of BPA may be compromised? Besides, could the authors explain the main improvements of BPA compared with LinBP?
4.	The paper primarily focuses on the enhancement of transfer-based attacks. Could you discuss potential applications or implications of the proposed BPA method in other areas of adversarial training, such as these gradient-based white-box attacks?

---

> ### Author Rebuttal · Authors · 2023-08-09
>
> We thank the reviewer for the insightful comments and suggestions. We address your concerns below.
>
> **Q.1**. Theoretical justification.
> >**A.1**. The theoretical explanation of transfer-based attacks is elusive, rendering the selection of specific modifications for boosting transferability a complex challenge. In this work, we find that the truncation of non-linear layers engenders a decline in the relevance between gradient w.r.t. input and loss function, which hampers adversarial transferability. Our approach endeavors to enhance the relevance by designing new derivatives for ReLU and max-pooling layers, which are non-zero and related to the magnitude of input. As shown in Fig. 4 (b), we have conducted parameter studies for max-pooling. To substantiate the rationale behind the modification for ReLU, we conduct empirical justification as follows:
> >
> >+ **Derivative Range Analysis**. We examine 1000 images and find that the majority of the elements in the derivative matrixs of ReLU fall within the interval $[-2,2]$ and more than $99.98\%$ elements are in the range $[-5,5]$. As shown in Fig. 3, the derivative of SiLU function is mainly different from that of ReLU function in the range of $[-5, 5]$. Hence, SiLU can work for almost all the elements.
> >+ **Scale the magnitude**. We add a scale factor $s$ for the derivative of SiLU to explore its amplitude influence. As shown in Tab. 6 (APDF), when increasing $s$, the attack performance firstly increases and achieves the peak at $s=0.8$. When we continue to increase the scale factor, it acts like LinBP which sets all the elements to ones and the performance starts to decay. Notably, the standard SiLU configuration ($s=1.0$) falls short of delivering optimal transferability. This underscores its untapped potential for further fortification of adversarial transferability.
> >
> >The empirical validations substantiates that while the chosen modifications exhibit promise in boosting transferability, the optimal choice may be yet undetermined. We will explore the potential modifications and try to derive theoretical justification in future work. Thanks.
>
> **Q.2**. Results on other datasets.
> >**A.2**. Existing transfer-based attacks mainly validate their effectiveness on ImageNet dataset. For a fair comparison, we also evaluate our BPA on ImageNet dataset. Here we conduct experiments on CIFAR-10 using PGD with $\epsilon=\frac{8}{255}$ on VGG-19. As shown in the table below, BPA exhibits better transferability than the baselines, showing its high effectiveness and generality to various datasets and models. We will add the complete results in the revision.
> >|Attacker|WRN|ResNeXt|DenseNet|pyramidnet|gdas|
> >|-|-|-|-|-|-|
> >|N/A|65.94|65.66|63.56|16.96|50.00|
> >|LinBP|67.88|67.46|65.02|18.18|51.78|
> >|Ghost|66.46|65.52|62.92|17.32|49.18|
> >|BPA|**74.38**|**73.80**|**69.66**|**20.74**|**57.16**|
>
> **Q.3**. Limitation and the main improvements of BPA.
> >**A.3**. Please refer to **G.3** for limitation. We summarize the main improvements of BPA compared with LinBP as follows:
> >+ **Motivation.** Different from LinBP which makes the deep model more linear, we find that the truncation of non-linear layers decays the relevance between the gradient w.r.t. input and loss function. It makes the gradient provide an imprecise direction to maximize the loss in some cases. This also explains why making the model more linear by LinBP results in better transferability.
> >+ **BPA considers non-linear layers.** Different from LinBP which only focuses on ReLU, we focus on recovering the gradient of non-linear layers, including ReLU and max-pooling. As shown in Table 5, with max-pooling, our BPA can achieve much better transferability than solely modifying ReLU, validating the effectiveness of considering a broader set of non-linear layers.
> >+ **BPA considers the magnitude of input.** Different from LinBP which uniformly sets all elements of ReLU's gradient to one, our BPA sets the gradient based on the magnitude of input matrix. It helps highlights the crucial elements based on the gradient, resulting in better transferability. To validate its advantage,  we compare the transferability of PGD on LinBP and BPA by solely changing the derivatives of ReLU. As shown in the following table, BPA exhibits better transferability than LinBP, supporting our argument.
> >
> >||Inc-v3|IncRes-v2|DenseNet|MobileNet|PNASNet|SENet|Inc-v3$_{ens3}$|Inc-v3$_{ens4}$|IncRes-v2$_{ens}$|
> >|-|-|-|-|-|-|-|-|-|-|
> >|LinBP|27.22|23.04|59.34|59.74|22.68|33.72|16.24|13.58|7.88|
> >|BPA|**29.38**|**24.00**|**62.80**|**61.82**|**24.98**|**34.96**|**17.52**|**14.38**|**8.90**|
> >
> >Though both LinBP and BPA modify the back-propagation process, they have distinct motivations and strategies to achieve better transferability. Considering a wider range of non-linear layers and modifying the gradient based on input magnitude, BPA achieves better adversarial transferability.
>
> **Q.4**. Potential applications of BPA.
> >**A.4**. As you suggested, we adopt FGSM to attack ResNet-50 with $\epsilon=\frac{8}{255}$ on ImageNet dataset. Integrating BPA with FGSM achieves a white-box attack success rate of $93.56\%$, which outperforms that of the vanilla model ($86.76\%$). This underscores the effectiveness of BPA for white-box attacks.
> >
> >Due to limited time and resources, we do not evaluate BPA for adversarial training. Smooth activation functions [1] can improve adversarial training, which shows the potential applicability of our BPA for adversarial training and warrants further exploration in future work.
> >
> >Note that this work mainly focuses on transfer-based attacks and does not discuss these potential applications in the paper. As you said, the proposed motivation and method might not be limited to transfer-based attacks. We will explore its potential application in adversarial learning in the future. Thanks for your insightful comments.
> >
> >[1] Xie et al. Smooth Adversarial Training. arXiv Preprint arXiv:2006.14536, 2020.

---

> > ### Comment · Reviewer_YaZN · 2023-08-13
> > **Response to authors' rebuttal**
> >
> > Thanks for the rebuttal. The response has addressed my major concerns, and I will raise my rating to 6.

---

### Official Review · Reviewer_g444 · 2023-07-05

**Soundness:** 2 fair
**Presentation:** 3 good
**Contribution:** 2 fair
**Rating:** 6
**Confidence:** 3

**Summary:**

This paper  have developed a new method, Backward Propagation Attack (BPA), to enhance the transferability of adversarial examples. BPA uses a non-monotonic function as the derivative of ReLU and incorporates softmax with temperature to smooth the derivative of max-pooling, reducing information loss during gradient backward propagation. The method has demonstrated significant improvements in adversarial transferability on the ImageNet dataset.

**Strengths:**

**Originality**: To the best of my knowledge, the idea that modifying the derivative of max-pooling, Relu activation for improving adversarial transferability is new.

 **Clarity**: This paper is well-structured and easy to follow.

**Significance**: To understand and improve the transferability of adversarial examples is important in the domain of adversarial examples

**Weaknesses:**

- The chosen baselines are exclusively drawn from model-related attacks. It might be beneficial to include a few from input-transformation-based attacks. Furthermore, it would provide a more comprehensive evaluation to examine if your technique could potentially work in conjunction with input-transformation-based attacks such as TIM[1]/SIN[2]/DIM[3] for further enhancing the transferability.

- There's a relevant piece of work that hasn't been included.[4]

[1]  Dong, Yinpeng, et al. "Evading defenses to transferable adversarial examples by translation-invariant attacks." Proceedings of the IEEE/CVF Conference on Computer Vision and Pattern Recognition. 2019.

[2] Lin, Jiadong, et al. "Nesterov accelerated gradient and scale invariance for adversarial attacks." arXiv preprint arXiv:1908.06281 (2019).

[3] Xie, Cihang, et al. "Improving transferability of adversarial examples with input diversity." Proceedings of the IEEE/CVF conference on computer vision and pattern recognition. 2019.

[4] Huang, Tianjin, et al. "Direction-aggregated attack for transferable adversarial examples." ACM Journal on Emerging Technologies in Computing Systems (JETC) 18.3 (2022): 1-22.

**Questions:**

- Could your technique potentially work in conjunction with input-transformation-based attacks such as TIM/SIN/DIM  for further improving the transferability?

**Limitations:**

the authors adequately addressed the limitations

---

> ### Author Rebuttal · Authors · 2023-08-09
>
> We thank the reviewer for the insightful comments and suggestions. We address your concerns below.
>
> **Q.1**. The chosen baselines are exclusively drawn from model-related attacks. It might be beneficial to include a few from input-transformation-based attacks. Furthermore, it would provide a more comprehensive evaluation to examine if your technique could potentially work in conjunction with input-transformation-based attacks such as TIM[1]/SIN[2]/DIM[3] for further enhancing the transferability.
>
> > **A.1**. In Table 1, we compare our BPA with various model-related attacks when combined with other attacks, including gradient-based (PGD), momentum-based (MI-FGSM, VMI-FGSM), objective-related (ILA) and input transformation based (SSA) attacks. Due to the page limit, we only evaluate BPA with one up-to-date transformation based attack (SSA). As you suggested, we further evaluate its generality to other input transformation based attacks, namely DIM, TIM and SIN in the following tables. As we can see, our BPA can significantly boost these input transformation based attacks. Compared with existing model-related attacks, BPA consistently exhibits better transferability for TIM and DIM. For SIN, BPA exhibits comparable attack performance with SGM on standardly trained models but much better transferability on adversarially trained models. To integrate the advantage of SGM and our BPA, we combine BPA with SGM for SIN, which outperforms the baselines with a clear margin. These results further validate its superiority in boosting adversarial transferability. We will add them in the revision. Thanks.
> >
> > **DIM**:
> >|Attacker|Inv-v3|InvRes-v2|DenseNet|MobileNet|PNASNet|SENet|Inc-v3$_{ens3}$|Inc-v3$_{ens4}$|IncRes-v2$_{ens}$ |
> >|-|-|-|-|-|-|-|-|-|- |
> >|N/A|45.00|38.56|71.64|70.08|41.60|48.56|28.52|24.82|16.48|
> >|SGM|52.72|45.22|79.42|82.34|49.50|58.66|32.42|28.20|19.26|
> >|LinBP|45.74|38.38|76.34|77.58|39.28|50.20|27.40|22.60|15.50|
> >|Ghost|45.20|37.86|72.70|72.34|40.32|48.50|27.44|23.98|15.78|
> >|BPA|**59.20**|**50.86**|**87.70**|**86.92**|**55.40**|**62.68**|**40.32**|**34.84**|**24.42**|
> >
> > **TIM**
> >|Attacker|Inv-v3|InvRes-v2|DenseNet|MobileNet|PNASNet|SENet|Inc-v3$_{ens3}$|Inc-v3$_{ens4}$|IncRes-v2$_{ens}$ |
> >|-|-|-|-|-|-|-|-|-|-|
> >|N/A|32.58|26.66|58.44|55.44|29.26|34.84|21.38|18.76|13.54|
> >|SGM|41.18|35.22|71.12|72.20|41.24|47.88|25.50|23.66|16.28|
> >|LinBP|43.20|36.30|75.08|74.08|39.14|46.90|27.62|23.62|16.74|
> >|Ghost|37.08|29.36|66.48|63.26|33.26|39.74|23.58|21.14|14.54|
> >|BPA|**59.20**|**50.86**|**87.70**|**86.92**|**55.40**|**62.68**|**40.32**|**34.84**|**24.42**|
> >
> > **SIN**
> >|Attacker|Inv-v3|InvRes-v2|DenseNet|MobileNet|PNASNet|SENet|Inc-v3$_{ens3}$|Inc-v3$_{ens4}$|IncRes-v2$_{ens}$ |
> >|-|-|-|-|-|-|-|-|-|-|
> >|N/A|42.68|33.76|70.12|65.70|34.90|39.90|26.36|23.48|15.22|
> >|SGM|52.78|43.04|79.92|80.22|45.10|52.70|31.94|27.44|18.74|
> >|LinBP|50.46|41.06|78.22|77.14|38.82|47.64|30.38|25.08|17.14|
> >|Ghost|47.46|37.92|77.14|72.82|38.66|46.02|29.14|24.58|16.28|
> >|BPA|52.40|43.44|80.12|76.38|45.42|50.28|39.16|36.32|26.06|
> >|SGM+BPA|**62.26**|**54.12**|**88.04**|**88.28**|**56.80**|**63.46**|**46.08**|**39.80**|**30.68**|
> >
> > [1] Dong, Yinpeng, et al. "Evading defenses to transferable adversarial examples by translation-invariant attacks." Proceedings of the IEEE/CVF Conference on Computer Vision and Pattern Recognition. 2019.
> >
> > [2] Lin, Jiadong, et al. "Nesterov accelerated gradient and scale invariance for adversarial attacks." arXiv preprint arXiv:1908.06281 (2019).
> >
> > [3] Xie, Cihang, et al. "Improving transferability of adversarial examples with input diversity." Proceedings of the IEEE/CVF conference on computer vision and pattern recognition. 2019.
>
> **Q.2**. There's a relevant piece of work that hasn't been included.[4]
> > **A.2** Direction-Aggregated adversarial attack (DA) utilizes the aggregated direction during the attack process to prevent the generated adversarial examples from overfitting to the white-box model. This approach aligns well with our BPA, making them compatible for combined evaluation. To further validate the effectiveness of our BPA, we integrate it with DA-MI-FGSM and present the results in the following table. The results demonstrate that BPA can significantly boost the adversarial transferability of DA-MI-FGSM, surpassing the performance of other model-related attacks. This further validates the superiority of BPA in boosting adversarial transferability with good generality to various attacks. We will add it in the revision. Thank you for your valuable feedback and contributions to strengthening our research.
> >|Attacker|Inv-v3|InvRes-v2|DenseNet|MobileNet|PNASNet|SENet|Inc-v3$_{ens3}$|Inc-v3$_{ens4}$|IncRes-v2$_{ens}$|
> >|-|-|-|-|-|-|-|-|-|-|
> >|N/A|45.00|38.56|71.64|70.08|41.60|48.56|28.52|24.82|16.48|
> >|SGM|52.72|45.22|79.42|82.34|49.50|58.66|32.42|28.20|19.26|
> >|LinBP|45.74|38.38|76.34|77.58|39.28|50.20|27.40|22.60|15.50|
> >|Ghost|45.20|37.86|72.70|72.34|40.32|48.50|27.44|23.98|15.78|
> >|BPA|**59.20**|**50.86**|**87.70**|**86.92**|**55.40**|**62.68**|**40.32**|**34.84**|**24.42**|
> >
> > [4] Huang, Tianjin, et al. "Direction-aggregated attack for transferable adversarial examples." ACM Journal on Emerging Technologies in Computing Systems (JETC) 18.3 (2022): 1-22.

---

> > ### Comment · Reviewer_g444 · 2023-08-11
> >
> > Thank you for the rebuttal. Most of my concerns have been addressed. Therefore,  I increase my score to 6.

---

> > > ### Author Response · Authors · 2023-08-12
> > > **A Gentle Reminder for Score Adjustment within the System**
> > >
> > > We are grateful for your positive comments and adjustment to the score. However, it appears that the score alteration has not been implemented within the system. **As a gentle reminder, we kindly request that you make the necessary score adjustment at your earliest convenience**. Thank you very much for your attention to this matter.

---

### Official Review · Reviewer_gcCC · 2023-07-06

**Soundness:** 4 excellent
**Presentation:** 4 excellent
**Contribution:** 4 excellent
**Rating:** 7
**Confidence:** 5

**Summary:**

In this work, the authors find that non-linear layers make the gradient imprecise, leading to limited adversarial transferability. Based on this finding, they propose Backward Propagation Attack (BPA) to increase such relevance. In particular, BPA adopts a non-monotonic function as the derivative of ReLU and incorporates softmax function to smooth the derivative of max-pooling. Extensive experiments have shown the high effectiveness of the proposed BPA.

**Strengths:**

1.The paper is well-written and easy to follow.
2.The motivation is clear and reasonable. Intuitively, the non-linear layers can indeed truncate the gradient during the backward propagation.
3.The proposed method is novel and interesting. Simply adjusting the gradient calculation of non-linear layers can significantly boost the adversarial transferability, which also validates the motivation.
4.Different from some works which is limited to some specific architectures, the proposed method is general to existing CNNs.
5.The authors have conducted experiments on both untargeted and targeted attacks, which can validate its high effectiveness and show good generality when combined with existing model-related methods.

**Weaknesses:**

1.For ReLU activation, the authors adopt the derivative of SiLU function. As shown in Figure 3, the derivative of SiLU function is mainly different from that of ReLU function in the range of [-5, 5]. I am curious how many elements the derivative matrix are in such range so that BPA can boost the transferability.
2.Recently, transforming the input image for gradient calculation has shown great effectiveness in boosting adversarial transferability, such as DIM, Admix, SSA and so on. It is expected to see the results on such attacks to further validate the effectiveness of BPA on various transfer-based attacks.

**Questions:**

1.What is the temperature coefficient for max-pooling?

---

> ### Author Rebuttal · Authors · 2023-08-09
>
> We thank the reviewer for the insightful comments and suggestions. We address your concerns below.
>
> **Q.1**. For ReLU activation, the authors adopt the derivative of SiLU function. As shown in Figure 3, the derivative of SiLU function is mainly different from that of ReLU function in the range of [-5, 5]. I am curious how many elements the derivative matrix are in such range so that BPA can boost the transferability.
>
> > **A.1**. As you suggested, we have examined the range of elements in the derivative matrix and observed that the majority of the elements fall within the interval $[-2,2]$. Additionally, more than $99.98\%$ elemnts are in this range $[-5,5]$. These findings strongly suggest that our gradient candidate possesses the ability to effectively handle most elements, making our BPA effective in boosting adversarial transferability.
>
> **Q.2**. It is expected to see the results on such attacks to further validate the effectiveness of BPA on various transfer-based attacks.
>
> > **A.2**. Our BPA modifies the backpropagation process, which is general to various transfer-based attacks. To validate its effectiveness, we adopt gradient-based (PGD), momentum-based (MI-FGSM, VMI-FGSM), objective-related (ILA) and input transformation based (SSA) attacks for evaluation. Due to the page limit, we only evaluate BPA with one up-to-date transformation based attack (SSA) in our paper. As you suggested, we further evaluate its generality to other input transformation based attacks in the following tables, namely DIM and TIM. As we can see, our BPA can significantly boost these input transformation based attacks and exhibits better transferability than other model-related works, which further validates its superiority in boosting adversarial transferability.
> >
> > **DIM**:
> > | Attacker | Inv-v3 | InvRes-v2 | DenseNet | MobileNet | PNASNet | SENet | Inc-v3$_{ens3}$ | Inc-v3$_{ens4}$ | IncRes-v2$_{ens}$ |
> > | -------- | ------ | --------- | -------- | --------- | ------- | ----- | --------------- | --------------- | ----------------- |
> > | N/A | 45.00 | 38.56 | 71.64 | 70.08 | 41.60 | 48.56 | 28.52 | 24.82 | 16.48             |
> > | SGM | 52.72 | 45.22 | 79.42 | 82.34 | 49.50 | 58.66 | 32.42 | 28.20 | 19.26 |
> > | LinBP | 45.74 | 38.38 | 76.34 | 77.58 | 39.28 | 50.20 | 27.40 | 22.60 | 15.50 |
> > | Ghost | 45.20 | 37.86 | 72.70 | 72.34 | 40.32 | 48.50 | 27.44 | 23.98 | 15.78             |
> > | BPA | **59.20** | **50.86** | **87.70** | **86.92** | **55.40**  | **62.68** | **40.32** | **34.84** | **24.42** |
> >
> > **TIM**:
> > | Attacker | Inv-v3 | InvRes-v2 | DenseNet | MobileNet | PNASNet | SENet | Inc-v3$_{ens3}$ | Inc-v3$_{ens4}$ | IncRes-v2$_{ens}$ |
> > | -------- | ------ | --------- | -------- | --------- | ------- | ----- | --------------- | --------------- | ----------------- |
> > | N/A      | 32.58  | 26.66     | 58.44    | 55.44     | 29.26   | 34.84 | 21.38           | 18.76           | 13.54             |
> > | SGM      | 41.18  | 35.22     | 71.12    | 72.20     | 41.24   | 47.88 | 25.50           | 23.66           | 16.28             |
> > | LinBP    | 43.20  | 36.30     | 75.08    | 74.08     | 39.14   | 46.90 | 27.62           | 23.62           | 16.74             |
> > | Ghost    | 37.08  | 29.36     | 66.48    | 63.26     | 33.26   | 39.74 | 23.58           | 21.14           | 14.54             |
> > | BPA      | **59.20**  | **50.86**     | **87.70**    | **86.92**     | **55.40**   | **62.68** | **40.32**           | **34.84**           | **24.42**             |
>
>
> **Q.3**. What is the temperature coefficient for max-pooling?
>
> >**A.3**. We set the temperature coefficient $t=10$ for ResNet-50 and conducted a parameter study in Section 4.4. We will add it into hyper-parameters in Section 4.1. Thanks.

---

### Official Review · Reviewer_NESJ · 2023-07-07

**Soundness:** 3 good
**Presentation:** 3 good
**Contribution:** 3 good
**Rating:** 7
**Confidence:** 3

**Summary:**

This paper introduces a new way to improve adversarial transferability by removing gradient truncation in the surrogate model. Specifically, the method replaces gradients of relu and maxpool with softened version to avoid gradient truncation. Experiments are conducted over many models and show gains upon existing methods.

**Strengths:**

- The method seems well motivated, novel and straightforward.
- The empirical gain is substantial and general across different settings
- The paper is well written overall and easy to follow.

**Weaknesses:**

It is not clear to me whether the improvement is caused by the extra design to align with the function better, or just simply by making the model more linear as suggested in LinBP. For the relu case, it seems that replacing gradient with 1 is the best as shown in prelimiary study. It is not clear whether this is still the case in the general experiments due to lack of ablation. For maxpool, Fig 2 also shows that the attack is best at some middle point. Does this relate to linearity too? And why does the success rate first increases then decreases?


**Questions:**

See above

**Limitations:**

Not addressed.

---

> ### Author Rebuttal · Authors · 2023-08-09
>
> We thank the reviewer for the insightful comments and suggestions. We address your concerns below.
>
> **Q1**. It is not clear to me whether the improvement is caused by the extra design to align with the function better, or just simply by making the model more linear as suggested in LinBP. For the ReLU case, it seems that replacing gradient with 1 is the best as shown in the preliminary study. It is not clear whether this is still the case in the general experiments due to the lack of ablation. For max-pooling, Fig 2 also shows that the attack is best at some middle point. Does this relate to linearity too? And why does the success rate first increase and then decrease?
>
> > **A.1**. 1) In this work, we find that the truncation of non-linear layers (e.g., ReLU, max-pooling) decays the relevance between the gradient w.r.t. the input and loss (see **G1** for more details). Making the model more linear can enhance such relevance, thus improving the transferability. However, making the model more linear is not optimal. Taking ReLU for example, we compare the transferability of PGD on LinBP and BPA by solely changing the derivatives of ReLU. Here LinBP makes the model more linear than our BPA. As shown in the following table, BPA exhibits better transferability than LinBP, which supports our argument.
> > | | Inc-v3 | IncRes-v2 | DenseNet | MobileNet | PNASNet | SENet | Inc-v3$_{ens3}$ | Inc-v3$_{ens4}$ | IncRes-v2$_{ens}$ |
> > |-|-|-|-|-|-|-|-|-|-|
> > |LinBP| 27.22 | 23.04 | 59.34 | 59.74 | 22.68 | 33.72 | 16.24 | 13.58 | 7.88|
> > |BPA| **29.38** | **24.00** | **62.80** | **61.82** | **24.98** | **34.96** | **17.52** | **14.38** | **8.90** |
> >
> > 2\) In the preliminary, we validate our hypothesis that we can increase such relevance to boost adversarial transferability. To increase the relevance, we randomly replace the zeros in the gradient of ReLU with ones. The increasing attack performance can validate our hypothesis but it does not yield an optimal solution.
> >
> > 3\) For max-pooling, the initial replacement of zeros in the gradient with ones results in an increase in relevance and subsequent improvement in attack performance. However, as the number of replaced zeros increases, the vallina BP faces challenges in accurately discerning the critical elements that are related to the magnitude of input values. This indiscriminate replacement introduces a substantial error, leading to a decay in attack performance. Hence, we adopt Equation (3) to address such issue. We also compare randomly replacing the zeros with ones using the probability of $0.3$ (the best results in Figure 2 (c)) and our BPA for max-pooling on ResNet-50. As shown in the following table, BPA exhibits better transferability, which further validates our motivation and indicates that the effectiveness of our BPA is not solely attributed to achieving linearity.
> >
> > | | Inc-v3 | IncRes-v2 | DenseNet | MobileNet | PNASNet | SENet | Inc-v3$_{ens3}$ | Inc-v3$_{ens4}$ | IncRes-v2$_{ens}$ |
> > |-|-|-|-|-|-|-|-|-|-|
> > |LinBP| 14.52 | 11.94 | 37.52 | 36.02 | 13.84 | 17.28| 10.34 | 10.56 | 6.48 |
> > |BPA| **20.26** | **16.16** | **44.66** | **42.82** | **17.12** | **21.52** | **13.20** | **11.88** | **7.74**|
> >
> > We will add the above results and discussion in the revision to clarify it. Thanks.

---

### Official Review · Reviewer_5iKz · 2023-07-08

**Soundness:** 3 good
**Presentation:** 3 good
**Contribution:** 2 fair
**Rating:** 5
**Confidence:** 4

**Summary:**

This paper introduces the Backward Propagation Attack (BPA) to improve the transferability of adversarial examples generated based on the iterative gradient ascent procedure. The paper first empirically demonstrates that the non-linear layers such as ReLU and max-pooling truncate the loss gradient, and such a truncation procedure negatively affects the transferability of adversarial perturbations. Motivated by this observation, the paper proposes to modify the gradient computation for the ReLU and max-pooling layers. Particularly, consider a trained source model from which the adversarial perturbations are generated. BPA proposes to replace the derivative of the ReLU functions in the source model with the derivative of the SiLU function. Similarly, the derivative of the max-pooling function in the source model is replaced with a softmax function with a tunable temperature coefficient. Finally, results on randomly sampled 5000 Imagenet images demonstrate the improved transferability of adversarial perturbations generated based on PGD and MI-FGSM.

**Strengths:**

Originality: Modifying the backward propagation procedure of the non-linear layers when computing the loss gradient during the process of generating adversarial examples is an interesting approach to improve the transferability of the perturbations.

Quality: The paper is well-written. The proposed BPA method is motivated using empirical observations. The evaluations include the transferability of perturbations under both targeted and untargeted settings.

Clarity: The motivation behind the proposed method as well as the structure of the paper is clear.

Significance: The adversarial robustness of deep learning models against transfer-based perturbations poses a practical security concern. This paper proposes a method that can be used in combination with existing gradient-based attack algorithms to improve the transferability of perturbations.


**Weaknesses:**

The clarity of the paper can be improved. There are some terms that are key to understanding the message which the author wants to convey, but the lack of rigorous definitions makes it difficult to understand. (See Questions for a list of clarifications )

The evaluations in the paper are performed based on perturbations bounded by the L_inf norm with an $\epsilon$ of $\frac{8}{255}$. In Ln 242, it states that the choice of $\epsilon$ aligns with existing work (which one?), whereas most existing work considers $\frac{16}{255}$ (LBP, MI, VI/VMI, etc). Especially since one of the main contributions of the proposed method is the improvement of transferability over other sota methods (ie. "remarkable margin" ln284, ln296, ln 358), it is important to make sure the evaluations and the comparison with other methods are performed fairly.

**Questions:**

Questions/clarifications:
In the abstract, why does the truncation make the gradient imprecise? The ReLU and the max-pooling layers are part of the network definition, so the derivative is exactly what it is supposed to be and I do not see the lack of precision. What does it mean by the relevance between gradient wrt input and the loss function? Since the term "relevance" appears later in the paper as well (eg. ln45, 58, 148, 315), having a more rigorous definition is warranted.

Figure 2 is a crucial element in motivating the proposed method. Is the success rate averaged over nine victim models? What is the variance among the results?

Ln 124 of Sec. 3.1: the paper first establishes that $z_i$ is the input to the function $f_i$ which returns $z_{i+1}$. When predicting the label, why is $z_l$ used as an input to the function $f_{l+1}$, shouldn't it be $z_{l+1}$?

Ln 138: Shouldnt the j-th element of $\frac{\partial z_{k+1}}{\partial z_k}$ be 1 if $f(z_{k})_j >0$?

Ln 139 "As a result, the gradient is effectively limited or weakened to some extent". ReLU was designed to address the vanishing gradient problem in training deep models. By having a step function as its derivative, it allows some gradient information to pass freely while simultaneously not updating other weights. Therefore, this is exactly how the backward propagation was designed to be with ReLU-activated networks.

Ln 414: Similarly, this is exactly how bp is supposed to behave with max-pooling.

Ln 149: What does it mean by "decaying the relevance"?

Ln 185-186: Is this a prior result? or a claim made in this paper supported by evidence?

Suggestions: Could we replace the y-axis ("Attack success rate") with the adversarial loss value on the source model? That is to plot $\ell(x+\Delta x)$, where $\Delta x$ is generated using fgsm/i-fgsm/mi-fsgm on the source model modified using the various method. If we observe a higher loss value after the modification, it indeed shows that the derivative of those nonlinear layers prevents the perturbations from reaching maxima. On the other hand, if we observe a lower loss value after the modification, this means the truncation effect of the derivative of the non-linear functions leads to perturbations overfit to the source model, despite having higher losses.

Other minor suggestions:
Figure 1 should not come after Figure 2.

**Limitations:**

There is no discussion of the limitation. For instance, the proposed paper is only suitable for attack algorithms that are based on gradient computations.

---

> ### Author Rebuttal · Authors · 2023-08-09
>
> We thank the reviewer for the insightful comments and suggestions. We address your concerns below.
>
> **Q.1**. Perturbation budget.
> >**A.1**. Both $\frac{8}{256}$ and $\frac{16}{256}$ are widely adopted perturbation budgets in adversarial learning. In this paper, we follow the setting of baselines [1,2] to set $\epsilon=\frac{8}{256}$ for fair comparison. As you said, there are many transfer-based attacks that have adopted $\frac{16}{256}$. Here we also evaluate BPA using PGD with $\epsilon=\frac{16}{256}$. As shown in Tab. 1 (APDF), BPA consistently outperforms the baselines with a clear margin, showing its remarkable effectiveness in boosting adversarial transferability. We will add the complete results in the revision.
> >
> >[1] Guo et al. Backpropagating Linearly Improves Transferability of Adversarial Examples. NeurIPS 2020.
> >
> >[2] Li et al. Learning Transferable Adversarial Examples via Ghost Networks. AAAI 2020.
>
> **Q.2**. Why does the truncation make the gradient imprecise?
> >**A.2**. Please refer to **G.1** for details. The relevance between gradient w.r.t. input and loss function indicates the sensitivity of loss function to changes in the input when taking a small step in the direction of gradient. This relevance can be formally defined as follows:
> >
> >**Definition 1 (Relevance between gradient w.r.t. input and loss function)** Given an input $x$, loss function $J(x)$ and a step size $\epsilon$, the relevance between gradient w.r.t. input and objective function can be defined as $\frac{J(x+\epsilon \cdot \nabla_xJ(x)) - J(x)}{\epsilon}$.
> >
> >As shown in **A.6**, BPA can increase the such relevance compared to vallina backpropagation, which further validates our motivation. We will clarify it in the revision. Thanks.
>
> **Q.3**. Is the success rate averaged over nine victim models in Fig. 2? What is the variance among the results?
> >**A.3**. Fig. 2 reports the average success rates on nine victim models due to page limit. As you asked, we calculate the variance of each data point in Tab. 2-4 (APDF). Since the attack success rates varies across different models, especially on standardly trained model (e,g., DenseNet) and adversarially trained model (e.g., Inc-v3$_{ens3}$), the large variance is expected and cannot provide any meaningful information. We also provide the original data for MI-FGSM of Fig. 2 (a) in Tab. 5 (APDF), which exhibit the consistent trend for each model. We will report these data in the appendix for reference.
>
> **Q.4**. Ln 124 & Ln 138.
> >**A.4**. They are typos. We will correct it in the revision.
>
> **Q.5**. Ln 139 "As a result, the gradient is effectively limited or weakened to some extent". ReLU was designed to address the vanishing gradient problem in training deep models. By having a step function as its derivative, it allows some gradient information to pass freely while simultaneously not updating other weights. Therefore, this is exactly how the backward propagation was designed to be with ReLU-activated networks. Ln 414: Similarly, this is exactly how bp is supposed to behave with max-pooling.
> >**A.5**. ReLU helps address the vanishing gradient issue during the training of deep models by eliminating some gradients (referred to as the truncation of ReLU in this paper). However, this paper focus on boosting transferability of adversarial examples generated on such ReLU-activated networks. To effectively generate adversarial examples, it is crucial that the gradient w.r.t. the input provides a reliable direction to maximize the loss. Unfortunately, as discussed in A.2, the truncation of ReLU makes the calculated gradient unnable to provide such exactly precise direction, i.e., making the gradient weakened to some extent. Similarly, max-pooling also truncates the gradient during the backpropagation, cuasing the gradient unnable to indicate an exactly precise direction.
>
> >In Fig. 2, we empirically validate that recovering such dropped gradient (though not precise) can effectively boost adversarial transferability. Motivated by this finding, we modify the backpropoagation for ReLu and max-pooling layers to recover the dropped gradient for better transferability.
>
> **Q.6**. Ln 149: What does it mean by "decaying the relevance"?
> >**A.6**. As responsed in **A.2** and **A.5**, the truncation of ReLU and max-pooling drops gradient in the backpropagation process, which decays the relevance. Based on Definition 1, we calculate the Relevance using vallina backpropagation and BPA on ResNet-50 with 1000 images. The gradient calculated by vallina backpropgation achieves the relevance of $149.23$, lower than BPA of $240.51$. This supports our hypothesis that non-linear layers can *decay the relevance*. We will add them in the Appendix and make it clearer.
>
> **Q.7**. Ln 185-186: Is this a prior result? or a claim made in this paper supported by evidence?
> >**A.7**. It is a claim made in this paper. As shown in Tab. 1 and Tab. 5, by only modifying ReLU, BPA achieves better performance than LinBP, supporting this claim. **G.2** further validates this claim.
>
> **Q.8**. Could we replace the y-axis ("Attack success rate") with the adversarial loss value on the source model?
> >**A.8**. In Fig. 2, we validate our hypothesis that we can increase such relevance to boost adversarial transferability. Thus, we adopt the Attack success rates as our y-axis. Since adversarial attacks aim to maximize the loss value to generate adversarial examples, there exists a positive relationship between the attack success rate and the average loss value. Following your suggestion, we have calculated the adversarial loss value, which shows a similar trend as attack success rate in Fig. 2. This analysis reinforces the findings drawn from the attack success rates, further supporting the hypothesis that increasing relevance leads to improved adversarial transferability.
>
> **Q.9**. Fig. 1 should not come after Fig. 2. & Limitation
> >**A.9**. We will swap their order. Please refer to **G.3** for limitation. Thanks.

---

> > ### Comment · Reviewer_5iKz · 2023-08-18
> >
> > Thank you for the response. Most of the concerns have been addressed and I will raise my score to 5.

---

### Author Rebuttal · Authors · 2023-08-09

We provide some experimental results in the attached PDF, denoted as APDF. We address the common concerns as follows:

> **G.1** If the gradient of a function is non-zero at a point $x$, the direction of gradient signifies the path in which the function experiences the steepest increase from $x$. Considering a function $f(x) = ReLU(wx+b)$ where $w >0, b, x \in R$, we have that $f(x)=0$ if $x\le-\frac{b}{w}$. Our goal is to maximize $f(x)$ starting from the data point $x_0$, which is analogous to adversarial attacks aimed to maximize the loss function. If $x_0 > -\frac{b}{w}$, $\frac{\partial f(x)}{\partial x}=w$ tells us that increasing $x$ will elevate $f(x)$. However, due to the truncation of ReLU, $\frac{\partial f(x)}{\partial x} =0$ when $x_0 \le -\frac{b}{w}$, fails to provide a clear direction for maximizing $f(x)$, rendering the optimization process ineffective. This truncation issue also arises during the backpropagation process in deep models, which we refer to as *"imprecise gradient"* in this paper. Thus, the *imprecise gradient* means that the gradient w.r.t. input image fails to indicate the direction to maximize/minimize the loss instead of the lack of precision for gradient calculation.

> **G.2** To validate that setting all elements in the corresponding derivative to ones is not optimal, we compare the transferability of PGD on LinBP and BPA by solely changing the derivatives of ReLU. As shown in the following table, BPA exhibits better transferability than LinBP, which validates our claim.
>| | Inc-v3 | IncRes-v2 | DenseNet | MobileNet | PNASNet | SENet | Inc-v3$_{ens3}$ | Inc-v3$_{ens4}$ | IncRes-v2$_{ens}$ |
>|-|-|-|-|-|-|-|-|-|-|
>|LinBP| 27.22 | 23.04 | 59.34 | 59.74 | 22.68 | 33.72 | 16.24 | 13.58 | 7.88|
>|BPA| **29.38** | **24.00** | **62.80** | **61.82** | **24.98** | **34.96** | **17.52** | **14.38** | **8.90** |

> **G.3** Limitation: Our proposed BPA modifies backpropagation process for gradient calcaulation, making it only suitable for gradient-based attacks. Besides, BPA modifies the derivatives of non-linear layers, such as ReLU and max-pooling. Consequently, it may not be directly applicable to models lacking these specific components, such as transformers. In the future, we will investigate how to generalize our BPA to such transformers by refining the derivatives of some components, e.g., softmax. This endeavor to enhance the generality and versatility of BPA will be an essential aspect of ongoing research, paving the way for the broader applicability of the proposed method and facilitating its adoption in various deep learning models beyond those with ReLU and max-pooling layers.
>
> We will add the limitation discussion in the revision.

---

### Decision · Program_Chairs · 2023-09-21

**Decision:**

Accept (poster)

**Comment:**

The paper presents a simple approach that improves the transferability of adversarial attacks. Based on the observation that the truncation of the gradients is highly correlated with the transferability of attack, the authors proposed to replace the gradient computations in ReLU and max-pooling to more smooth functions, and demonstrated the consistent improvement in attack success rates.

Overall, all reviewers were satisfied with the clarity and empirical validation of the approach. The paper initially received two Accepts, two Borderline Accepts, and one Borderline Reject. Primary concerns raised by the reviewers were about comparisons with various transfer attacks (e.g., the ones based on input transformation), comparisons on more datasets, clarifications on some claims (e.g., is linearity enough to enhance the relevance of the input to the loss?) and terminology (e.g., term “relevance”). The authors adequately addressed most concerns in the rebuttal, hence all reviewers recommend acceptance. The AC agrees with the reviewers’ decision. The authors should update the camera-ready paper to include the changes suggested by the reviewers during the review process.